# The Speciated Isoprene Emission Model with the MEGAN Algorithm for China (SieMAC)

Shengjun Xi<sup>1</sup>, Yuhang Wang<sup>1</sup>, Xiangyang Yuan<sup>2</sup>, Zhaozhong Feng<sup>2</sup>, Fanghe Zhao<sup>1</sup>, Yanli Zhang<sup>3</sup>, Xinming Wang<sup>3</sup>,

- 5 School of Earth and Atmospheric Sciences, Georgia Institute of Technology, Atlanta, 30318, United States
  - <sup>2</sup> Key Laboratory of Ecosystem Carbon Source and Sink, China Meteorological Administration (ECSS-CMA), School of Ecology and Applied Meteorology, Nanjing University of Information Science & Technology, Nanjing, 210044, China
    <sup>3</sup> State Key Laboratory of Organic Geochemistry, Guangzhou Institute of Geochemistry, Chinese Academy of Sciences, Guangzhou, 510640, China
- 10 Correspondence to: Yuhang Wang (yuhang.wang@eas.gatech.edu)

Abstract. Isoprene is the dominant non-methane Volatile Organic Compound (VOC) emitted from terrestrial ecosystems and plays an important role in ozone chemistry. Understanding isoprene emissions is critical for controlling air pollution. The Model of Emissions of Gases and Aerosols from Nature (MEGAN) is widely used to calculate biogenic isoprene emissions worldwide. While MEGAN predictions are good for many regions, a previous analysis of isoprene observations around China showed large discrepancies between observed and simulated isoprene concentrations. The uncertainties of isoprene emissions in China are also reflected in the large differences between MEGAN version 2.1 and 3.1. In this work, bottom-up high-resolution vegetation distributions and updated emission factors are combined with satellite data in the Speciated Isoprene Emission Model with the MEGAN Algorithm for China (SieMAC) to improve isoprene emission estimates in China. The results from this new emission inventory for summer 2013 improve upon MEGAN versions 2.1 and 3.1 when compared with isoprene observations and satellite HCHO products. This improved emission inventory is applied in a regional model, and the results indicate a potentially underestimated role of biogenic isoprene in ozone formation over polluted eastern China.

#### 1 Introduction

Isoprene (C<sub>3</sub>H<sub>8</sub>) dominates global biogenic volatile organic compound (BVOC) emissions, accounting for approximately half of total BVOC emissions and one-third of all volatile organic compounds (VOCs) released into the atmosphere (Guenther et al., 2012). Its high reactivity makes it a crucial precursor for tropospheric ozone (O<sub>3</sub>) formation, particularly in regions with substantial biogenic sources (Fiore et al., 2005; Fu et al., 2007; Geng et al., 2011; Fu and Tai, 2015; Zhang and Wang, 2016; Ma et al., 2019; Li et al., 2019a; Wu et al., 2020; Wang et al., 2021b; Geddes et al., 2022; Lou et al., 2023; Oumami et al., 2024).

This is especially significant in China, where isoprene comprises over 50% of annual BVOC emissions and surpasses anthropogenic VOC levels during summer daytime hours when ozone production peaks (Tie et al., 2006; Li et al., 2016; Wang et al., 2020). Recent studies show that isoprene can enhance regional ozone production rates by up to 10 ppbV h<sup>-1</sup> during summer (Geng et al., 2011; Wu et al., 2020; Yu et al., 2022). The importance of accurately quantifying isoprene emissions has

Deleted: s

Deleted: s

grown as China faces persistent and severe summertime ozone pollution (Li et al., 2019c; Lu et al., 2020; Li et al., 2020b; Wang et al., 2023). This highlights the critical need for well-constrained isoprene emission estimates to better understand regional atmospheric chemistry and develop effective air quality policies.

The Model of Emissions of Gases and Aerosols from Nature (MEGAN) is the most widely used biogenic emission model and serves as the standard module for estimating biogenic isoprene emissions in chemistry transport models. However, MEGAN-based national isoprene emission estimates for China ranged from 4.10 to 37.45 TgC yr. (Guenther et al., 1995; Klinger et al., 2002; Tie et al., 2006; Chi and Xie, 2012; Fu and Liao, 2012; Li et al., 2012; Li et al., 2013; Li and Xie, 2014; Stavrakou et al., 2014; Gao et al., 2019; Li et al., 2020c; Wang et al., 2020; Wu et al., 2020; Wang et al., 2021; Ma et al., 2022; Li et al., 2023) and regional studies showed uncertainties between -82% and +177% (Zheng et al., 2010; Wang et al., 2011b). Furthermore, a previous analysis of isoprene observations around China showed large discrepancies between observed isoprene concentrations and those simulated with MEGAN emission inventories (Zhang et al., 2020).

45

65

These uncertainties in MEGAN are largely attributed to its Emission Factor (EF) input (Wang et al., 2011b; Situ et al., 2014). EF represents the emission potential of plants, and ranges from 0 to >10000  $\mu$ g m<sup>-2</sup> h<sup>-1</sup> among plant species, with substantial variations even within individual genera (Klinger et al., 2002; Guenther et al., 2006; Guenther et al., 2012). Ideally, to generate precise EF inputs, species-specific EF measurements and species-level plant composition data are required. Unfortunately, such data specificity is not available for some regions (Messina et al., 2016; Guenther et al., 2006). MEGAN provides a global gridded EF map with 1 km spatial resolution as the default input (Guenther et al., 2012; Chen et al., 2022). While this high-resolution map incorporates speciated EF data for regions with adequate measurements and detailed vegetation distributions, the values assigned to most regions in China were based on an outdated ecoregion map from and global ecoregion-average EFs (Guenther et al., 2006; Olson et al., 2001), failing to capture the country's diverse vegetation composition and recent land cover changes (Peng et al., 2014; Chen et al., 2019). Although MEGAN version 3.1 incorporates more recent EF measurements for China, it still relies on the previous ecoregion distributions (Jiang et al., 2018b; Chen et al., 2022).

Recent studies have begun utilizing local EF measurements and plant species distribution datasets to drive MEGAN for China (Wang et al., 2011b; Li et al., 2013; Situ et al., 2013; Li and Xie, 2014; Situ et al., 2014; Wang et al., 2018; Li et al., 2020c). However, few of these estimates are evaluated against observational data, and those incorporating both local EFs and detailed vegetation distributions are largely confined to regional scales. For example, Li et al. (2020c) investigated national-scale isoprene emissions using local EFs, but their reliance on outdated vegetation distributions limits the accuracy of their estimates. To our knowledge, no comprehensive national study has estimated isoprene emissions across China by integrating both localized EFs and updated vegetation distribution, highlighting a critical gap in the field that needs to be addressed.

To address these challenges, we present the Speciated Isoprene Emission Model with the MEGAN Algorithm for China (SieMAC). SieMAC builds upon the MEGAN framework with four key updates: (1) incorporation of extensive local EF measurements and up-to-date vegetation distributions, (2) implementation of Plant Functional Type (PFT)-specific Leaf

Deleted:

Formatted: Superscript

Area Index (LAI), (3) addition of an optional environmental factor accounting for water-stress impacts, (4) modification of the temperature response algorithm for boreal grass.

We implement SieMAC into a three-dimensional regional chemical transport model (REAM) and evaluate its performance using both ground-based measurements and satellite observations. The evaluation results demonstrate that SieMAC outperforms both MEGAN version 2.1 and MEGAN version 3.1 in representing isoprene emissions across China. The paper describes the model algorithms and input datasets (Section 2), evaluation methodology (Section 3), results and validation (Section 4), and discusses uncertainties and implications for understanding regional air quality (Section 5).

## 2 Model Description

80

90

## 2.1 SieMAC model algorithm

SieMAC adapts the MEGAN algorithm to calculate vegetation isoprene emissions. The model is based on MEGAN v2.1's offline version (available at <a href="https://bai.ess.uci.edu/megan">https://bai.ess.uci.edu/megan</a>, last access: 6 Sept 2025) and calculates emissions as:

$$Emission = EF \cdot \gamma \tag{1}$$

where EF is the emission factor representing emissions under the standard condition as defined in Guenther et al. (2012) and  $\gamma$  is the normalized emission activity factor accounting for deviations from the standard condition.

MEGAN provides a global 1km × 1km EF map as the default input for emission factor, while also supporting PFT-specific emission factors as an alternative option (Guenther et al., 2006; Guenther et al., 2012; Oumami et al., 2024). SieMAC adopts the PFT-specific approach, which allows dynamic adjustment of EF inputs for different regions and periods. Following the PFT-7 classification scheme (Table 1), the model calculates grid cell emission factors as:

$$EF = \sum_{i} F_{i} \cdot EF_{i} \tag{2}$$

where  $F_i$  is the fraction of grid area covered by PFT i, and  $EF_i$  is its specific EF based on plant composition of the PFT.

The emission activity factor  $\gamma$  accounts for environmental influences on emissions:

$$\gamma = (\sum_{canopy\ layer\ =\ l} w_l \cdot \gamma_{P,l} \cdot \gamma_{T,l}) \cdot \gamma_A \cdot \gamma_{SM} \cdot \gamma_{CO_2} \cdot LAI \cdot C_{CE}$$
(3)

where the components represent effects of light ( $\gamma_P$ ), temperature ( $\gamma_T$ ), leaf age ( $\gamma_A$ ), soil moisture ( $\gamma_{SM}$ ), CO<sub>2</sub> inhibition ( $\gamma_{CO_2}$ ), and leaf area index (*LAI*).  $C_{CE}$  is a unitless constant that normalizes  $\gamma$  to unity under the standard condition. SieMAC implements MEGAN's standard canopy environment model, which simulates light and temperature distributions at five canopy depths. The overall impact of light and temperature ( $\gamma_P \cdot \gamma_T$ ) is calculated as the weighted average of these factors across the canopy layers using the model's predefined layer-specific weights ( $w_I$ ). In addition, we modify the calculation of  $\gamma_T$  for boreal grass following Wang et al. (2024b), and the affected regions are shown in Fig. S1. The calculation of  $\gamma_A$  follows MEGAN v2.1's default algorithm. The values of  $\gamma_{SM}$  and  $\gamma_{CO_2}$  are set to 1 due to uncertainties in characterizing these processes (Situ et al., 2014; Jiang et al., 2018b; Wang et al., 2021b; Pang et al., 2024).

A key enhancement in SieMAC is the use of PFT-specific LAI rather than grid cell averaged values, as LAI can vary substantially among different vegetation types (e.g., broadleaf trees typically have greater leaf area than grasses) and show distinct seasonal patterns (Oleson and Bonan, 2000; Bonan et al., 2002). Additionally, SieMAC includes an optional factor,  $\gamma_{VPD}$ , to account for enhanced isoprene emissions under water-stressed conditions, addressing a limitation in MEGAN v2.1 where soil moisture effects ( $\gamma_{SM}$ ) are not fully characterized (Zhang and Wang, 2016; Ma et al., 2019).  $\gamma_{VPD}$  is calculated based on the vapor pressure deficit (VPD), defined as the difference between saturation vapor pressure and ambient vapor pressure (Zhang and Wang, 2016). VPD is directly related to water stress of plants, with higher VPD values indicate more intense water-stressed conditions. While  $\gamma_{VPD}$  was derived from U.S. observations and its application to other regions requires careful evaluation, it provides a mechanism to investigate water-stress impacts on isoprene emissions in China. The final emission calculation in SieMAC is:

$$Emission = (\sum_{canopy\ layer\ =\ l} w_l \cdot \gamma_{P,l} \cdot \gamma_{T,l}) \cdot \gamma_A \cdot \gamma_{VPD} \cdot C_{CE} \cdot \sum_i F_i \cdot EF_i \cdot LAI_i \tag{4}$$

where i is the index of PFT, and  $\gamma_{VPD}$  is optional. Details of datasets and methods used to derive driving variables for SieMAC are described below.

#### 2.2 SieMAC land cover variables

SieMAC requires two categories of driving variables: weather variables (light, temperature, wind speed, humidity and pressure) and land cover inputs. For land cover inputs, we develop two datasets at different spatial resolutions: a High Resolution (HR) dataset and a Moderate Resolution (MR) dataset. These datasets include PFT fractions ( $F_i$  in Eq. (2)), PFT-specific emission factors ( $EF_i$  in Eq. (2)), and leaf area indices ( $LAI_i$  in Eq. (2)). Given that unrepresentative land cover inputs can significantly bias emission estimates, we derive these datasets through careful integration of multiple data sources. We use these variables to estimate isoprene emissions across China during summer 2013 and evaluate the results using previous studies, satellite products, and ambient measurements.

#### 120 2.2.1 Fraction and emission factor

125

We first calculate the fraction of a grid cell area covered by each PFT and their corresponding emission factors (hereafter PFT F and PFT EF, respectively), and then use these results to derive PFT-specific LAIs for each grid. Three datasets are used for this derivation: the Vegetation Atlas of China (1:1000000) (Zhang, 2007), eighth China Forest Resource Statistics (https://www.forestdata.cn/, last access: 15 May 2025), and MODIS MOD44B v006 product for 2013 (https://lpdaac.usgs.gov/products/mod44bv006/, last access: 15 May 2025, Dimiceli et al., 2015).

Our land cover analysis integrates these three complementary datasets to provide a complete picture of vegetation distribution in China. The Vegetation Atlas of China (1:1000000) serves as our baseline for vegetation composition, providing species-level distributions based on nationwide surveys from the 2000s (Li et al., 2013; Li and Xie, 2014). However, this information needs to be updated. We therefore incorporate the eighth Forest Resource Statistics, a five-year national survey

that provides genus-level forest composition data at the provincial level for 2009-2013 and the MODIS MOD44B v006 product, which provides annual tree and non-tree vegetation areas at a 250 m resolution for 2013.

135

140

155

160

Given that no single dataset provides up-to-date species-level vegetation distribution for China, we develop two processing approaches – High Resolution (HR) and Moderate Resolution (MR) - to optimize the use of available information. The HR approach prioritizes detailed species composition from the Vegetation Atlas while adjusting tree areas using the Forest Resource Statistics, whereas the MR approach directly applies provincial-level tree compositions to MODIS tree area data. HR provides more accurate species composition data, but it is based on an older national survey, while MR takes advantage of concurrent satellite tree cover data with provincial-level species composition constraints, as shown in Fig. 1. The detailed data integration workflows for both approaches are further illustrated in Figures S2 (MR) and S3 (HR). For both datasets, we first disaggregate the genus-level tree compositions from the Forest Resource Statistics into species-level information using the species proportion data from the Vegetation Atlas. This step provides provincial-level species composition data that are consistent with the two datasets.

For the HR dataset related to trees, we compare provincial-level areas between the Forest Resource Statistics and the Vegetation Atlas for each tree PFT. When the Atlas shows larger areas, we proportionally scale down all species areas within that PFT to match the Forest Resource Statistics while maintaining relative species proportions. Conversely, when the Forest Resource Statistics show larger areas, we preserve the Atlas species distributions and calculate residual areas. These residuals are then redistributed across grid cells where MODIS indicates tree cover after accounting for Atlas-based distributions. To implement this redistribution, we regrid both the Vegetation Atlas ( $\sim$ 10km²) and MODIS tree products to a 4× 4 km² resolution, ensuring that the total areas for all tree PFTs match the Forest Resource Statistics

The MR dataset for trees takes a different approach by directly applying the derived provincial species compositions to MODIS tree area data at the 250 m resolution. This assumes uniform tree species composition within each province but captures concurrent vegetation cover more accurately. The resulting fractions are then averaged to 500 m grid cells to match the resolution of LAI data to be described in the following section.

For non-tree PFTs (shrubs, crops, and grass), both HR and MR datasets use the same approach. We combine MODIS non-tree vegetation areas with PFT type from the Vegetation Atlas. For grid cells with MODIS non-tree vegetation coverage but lacking non-tree vegetation classification in the Atlas, we assign the non-tree PFT types based on the nearest classified grid cell. We calculate PFT fractions with these PFT areas, which are then averaged to a 4 km resolution for HR and a 500 m resolution for MR datasets.

To derive PFT EFs, we compile leaf-level measurements from published studies for tree and shrub species (Guenther et al., 1995; Yang et al., 2001; Klinger et al., 2002; Li and Klinger, 2002; Wang et al., 2002; Zhang et al., 2002; Wang et al., 2003a; Wang et al., 2003b; Zhao et al., 2004; Kang et al., 2005; Luo et al., 2005; Geron et al., 2006; Wang et al., 2007; Singh et al., 2008; Chen et al., 2009; Dong et al., 2009; Tsui et al., 2009; Zhang and Xie, 2009; Deng et al., 2010; Yin et al., 2010; Huang et al., 2011; Li and Xiang, 2011; Wang et al., 2011a; Zhu, 2011; Song et al., 2012; Yang et al., 2013; Li et al., 2014; Situ et al., 2014; Bao, 2015; Gao, 2016; Tang et al., 2016; Wang et al., 2016; Bu et al., 2017; Li et al., 2017; Lin et al., 2017;

| Deleted:   |  |
|------------|--|
| Deleted: ( |  |
| Deleted: ) |  |

Wang et al., 2017b; Chen et al., 2018; Jiang et al., 2018a; Jin et al., 2018; Pan et al., 2018; Wang et al., 2018; Li et al., 2019b; Li et al., 2019d; Li et al., 2019; Li et al., 2020a; Liu et al., 2020; Peng et al., 2020; Pang et al., 2021), as they contribute most significantly to isoprene emissions and show high variability in emission capacities (Guenther et al., 2012). For species lacking direct measurements, we assign EFs based on genus-average values. When genus-average values are unavailable, we adopt default values from Guenther et al. (1995), Wang et al. (2003a), and Wang et al. (2016). For crops and grasses, we use MEGAN v2.1 default EFs (Guenther et al., 2012) due to limited measurements and generally lower emission rates. However, based on recent findings of higher emission potentials in boreal regions (Wang et al., 2024b), we increase the grass EFs by a factor of three in northwestern provinces of China (Fig. S1), where sedge comprises a significant portion of the grass coverage.

#### 175 2.2.2 Leaf area index

185

190

We make use of MODIS measurements in conjunction with the high-resolution PFT distribution described in the previous section to derive LAI distributions. The three MODIS products include MCD15A2H (https://lpdaac.usgs.gov/products/mcd15a2hv006/, last access: 15 May 2025, Myneni et al., 2015), MCD12Q1 (https://lpdaac.usgs.gov/products/mcd12q1v006/, last access 15 May 2025, Friedl and Sulla-Menashe, 2019), and MOD44B. MCD15A2H is MODIS LAI product, delivering mean LAI values for each pixel on a 8-day basis. MCD12Q1 is MODIS land cover product, providing global maps of land cover with five legacy classification schemes at annual time steps; we apply the PFT classification scheme land cover dataset for 2013. Both MCD15A2H and MCD12Q1 have a spatial resolution of 500 m, while MOD44B has a resolution of 250 m. We aggregate MOD44B data to a 500-m resolution to be compatible with the other MODIS datasets. A key issue is to properly account for the differences between the Vegetation Atlas and MODIS PFT product since the former can be used to assign species-specific isoprene EFs (Table S1), but the latter cannot. Additionally, the derivation of MODIS LAI product incorporates its land cover product as a priori data (Knyazikhin et al., 1999). To accommodate these incompatibilities, we first define eight main ecoregions (Fig. S4, RodrÍguez and PÉrez, 2013; Zhang, 2007). An ecoregion is a geographical area that has a similar set of ecosystems. As such, a given PFT in MODIS can represent a different ecosystem set in a different ecoregion. It should be noted that the eight ecoregions are used exclusively for the purpose of calculating LAI data. We therefore compute 56 PFT-ecoregion dependent base LAIs using MODIS data (Table S2). In the second step, we apportion grid-cell MODIS LAI data into PFT-specific LAIs to maintain the spatial variability observed by MODIS.

We first compute monthly averaged PFT-ecoregion dependent base LAIs. The MODIS MCD12Q1 product only assigns one dominant PFT type to each 500 m grid cell, and we apply MOD44B vegetation coverage data to determine the fraction of the assigned PFT in each grid cell. We compute the monthly PFT-ecoregion dependent LAI data for PFT *i* and ecoregion *j*:

$$LAI_{i,j} = \frac{\sum_{k=1}^{k=N_j} {}_{LAI_k}}{\sum_{k=N_j} {}_{F_k}}$$
 (5)

Deleted: S2

where  $N_j$  is the number of grid cells with PFT i in ecoregion j,  $LAI_k$  denotes MODIS LAI value for grid cell k, and  $F_k$  denotes 200 PFT i fraction in grid cell k.

We then add MODIS observed grid cell level LAI variability to the PFT-ecoregion average LAIs. MODIS LAI measurement gives the average cell  $LAI_{MODIS}^c$ , which is the sum of area-weighted LAI for all PFTs in the grid cell:

$$LAI_{MODIS}^{c} = \sum_{i} \alpha_{i} \cdot F_{i}^{c} \cdot LAI_{i,i} \tag{6}$$

where  $\alpha_i$  accounts for the discrepancies between the PFT area fractions,  $F_i^c$ , obtained using the Vegetation Atlas of China, China Forest Resource Statistics, and MODIS MOD44B v006 product as described in the previous section, and those applied by MCD15A2H as prior information. We assume that  $\alpha_i$  does not vary with PFT, therefore:

$$\alpha = \frac{LAI_{MODIS}^{G}}{\sum_{i} F_{i}^{C} \cdot LAI_{i,j}} \tag{7}$$

For each grid cell and PFT i, we then constrain the LAI value as:

$$LAI_{i}^{c} = \alpha \cdot LAI_{i,i} \tag{8}$$

The LAI<sub>i</sub><sup>c</sup> represents the LAI value for PFT i in grid cell that can be directly used in calculating the emissions from the PFT, which is the PFT LAI input required for SieMAC. The 500 m PFT LAI data are directly used for MR emissions and aggregated to a 4 km resolution for HR emissions. We limit the range of the resulted PFT LAI values following Bonan et al. (2002). For grid cells lacking valid MODIS LAI values, which occurs when MODIS PFT product classifies a grid as non-vegetation despite the presence of some vegetation cover (Wang et al., 2018; Wang et al., 2021a), we assign PFT-ecoregion average LAIs as default values. This approach allows us to account for vegetation contributions that might otherwise be missed in grid cells dominated by non-vegetation surfaces. The resulting spatial distributions of PFT-specific LAIs and emission factors are shown in Figures S5-S8.

## 3 Model Evaluations

#### 3.1 Model Setup

We implemented SieMAC into REAM to evaluate emission estimates and simulated ambient isoprene levels against measurements. REAM is a three-dimensional regional chemistry transport model that has been applied in numerous tropospheric chemistry and transport studies across the United States and China (Zhao et al., 2009; Zhao and Wang, 2009; Zhao et al., 2010; Zhang and Wang, 2016; Zhang et al., 2018; Li et al., 2019a; Qu et al., 2020; Zhang et al., 2020; Yan et al., 2021). The overall model setup and SieMAC integration approach are illustrated in Figure S9. The model has a spatial resolution of 36 × 36 km² and includes 30 vertical layers in the troposphere. Meteorological fields are obtained from the Weather Research and Forecast (WRF) model V4.0 simulations, constrained by ECMWF Reanalysis v5 (ERA5) products. The chemical mechanism is based on GEOS-Chem (v11.01) with extended aromatic reactions from SAPRC-07. Lateral boundary conditions are obtained from a 2° × 2.5° GEOS-Chem model (v11.01) simulation. For anthropogenic emissions, we

Formatted: Heading 2

used the Multi-resolution Emission Inventory for China (MEIC) for 2013, while biogenic emissions for all chemical species except isoprene are calculated using MEGAN v2.1. We selected summer 2013 for this study as it represents the first year with concurrent availability of CARE-China isoprene measurements and the China National Environmental Monitoring Centre (CNEMC) Network air quality observations, enabling comprehensive model evaluation (Zhang et al., 2020; Bai et al., 2020).

We conducted model simulations for summer 2013 using six different isoprene emission configurations as detailed in Table 2. For the MEGAN v2.1 and v3.1 simulations, we utilized their respective offline versions and EF inputs available at https://bai.ess.uci.edu/megan/data-and-code. It needs to be noted that MEGAN v3.1 EFs are provided with different J-ratings, a measure of confidence for corresponding EF values. Here, we used the EF values with a J-rating value of 4, the highest confidence, to obtain the most accurate EF inputs for MEGAN v3.1. We followed the procedure in Wang et al. (2021a) to obtain LAI data (denoted as LAIv in that study) as well as PFT distributions. We averaged the LAIs, PFT fractions, and MEGAN 1 km EFs at each REAM grid cell and used the results as land cover inputs for MEGAN models. Note that MEGAN v3.1 results should be considered preliminary as the developers are currently finalizing MEGAN v3.2.

The SieMAC implementation in REAM involved a multi-scale integration approach. We first simulated the canopy environment for each REAM grid cell with local weather variables, including hourly temperature, solar radiation, humidity, wind speed and pressure derived from the WRF simulations, and grid cell-averaged LAI and PFT fractions derived from the high-resolution land cover datasets, using the MEGAN canopy model. For simulations using the MR dataset, the REAM grid-averaged LAI and PFT fractions were calculated by aggregating the 500 m resolution data, while for HR simulations these parameters were derived from the 4 km resolution datasets. The environmental activity factors calculated for each REAM grid cell were then applied to their corresponding high-resolution subgrid cells. Emissions were first calculated at finer resolutions (500 m for MR and 4 km for HR), then aggregated to the REAM resolution by summing emissions from all subgrid cells within each REAM cell.

## 3.2 CARE-China Observations

The Campaign on Atmospheric Aerosol Research Network of China (CARE-China) collected ambient air samples at twenty sites across China (Fig. 2) at approximately 14:00 Beijing Time every Wednesday from March 2012 to April 2014, providing valuable isoprene measurements for model-measurement comparison. Details about CARE-China observations are described by Zhang et al. (2020). We compared REAM simulated isoprene concentrations using different emission inventories with these CARE-China observations to evaluate the model performance. We analysed the model performance across six regions: North China Plain (NCP), Northeast China Plain (NECP), Northwest China (NWC), Southwest China (SWC), Southeast China (SEC), and South China (SC). The Lhasa site was excluded from analysis due to limited observations during the study period.

Deleted: 1

Formaldehyde (HCHO) serves as an intermediate product of isoprene oxidation and provides a valuable proxy for evaluating isoprene emissions. During summer, when isoprene emissions peak and dominate reactive VOC emissions, HCHO spatial variability closely aligns with isoprene emission patterns (Palmer et al., 2003; Shim et al., 2005; Millet et al., 2008). We evaluated the spatial distribution of different emission inventories obtained in this study using the monthly Level 3 OMI HCHO Vertical Column (VC) product with a 0.25 ° × 0.25 ° spatial resolution, from the Belgian Institute for Space Aeronomy (BIRA-IASB) (De Smedt et al., 2012; De Smedt et al., 2015), increasing observed concentrations by 32% to correct the low bias of OMI HCHO products for high HCHO regions as suggested by Zhu et al. (2020).

#### 4 Results and Discussion

#### 4.1 Model Evaluations

#### 4.1.1 Comparison with other studies

The national isoprene emissions estimated by SieMAC MR and HR for summer 2013 are 10.92 and 11.37 Tg C, respectively (Table 3). Based on the seasonal distribution pattern where summer accounts for approximately 70% of annual emissions (Tie et al., 2006; Li et al., 2023), these estimates correspond to annual emissions of around 15.6-16.2 Tg C. These values fall within the range of previous estimates and notably align with the 15 Tg C annual estimate reported by Guenther et al. (1995). However, the SieMAC emission distributions differ from previous studies, which will be elaborated in section 4.2.

The SieMAC HR estimate is approximately 4.1% higher than the MR estimate, reflecting their different approaches to characterizing tree distributions. In HR, tree areas and distributions are constrained by the Forest Resource Statistics and the Vegetation Atlas, respectively. In contrast, MR relies on MODIS products for tree area and assumes uniform tree composition at the provincial level. The impact of these methodological differences is well illustrated in Heilongjiang province, where the tree coverage is 1.95×10<sup>5</sup> km² in HR compared to 1.49×10<sup>5</sup> km² in MR. This difference in total tree area, approximately 31% higher in HR, directly explains the 32% higher emission estimate given by the HR approach for this province.

Comparing SieMAC with MEGAN simulation results using their respective default emission factor maps, REAM grid-average LAI and PFT fractions, and identical meteorological inputs, we find that SieMAC HR and MR estimates exceed both MEGAN v2.1 (8.05 TgC) and MEGAN v3.1 (4.47 TgC) predictions for summer 2013. Four key factors contribute to SieMAC's higher estimates: (1) the implementation of PFT-specific LAI, which better captures emissions from high-emitting vegetation types (Messina et al., 2016); (2) incorporation of China's recent afforestation, which has doubled national forest cover (Zhang et al., 2016; Chen et al., 2019); (3) improved representation of mixed vegetation areas, addressing a MODIS LAI retrieval limitation that can reduce emission estimates by up to 19% (Wang et al., 2018); and (4) updated emission factors for shrub species that are significant isoprene emitters in China (Klinger et al., 2002).

Including VPD effects in SieMAC increases emission estimates by approximately 40%, pushing estimates toward the upper range of previous studies. While this VPD algorithm was developed based on the observations in the United States (Beckett et al., 2012; Zhang and Wang, 2016), and its direct application to Chinese vegetation introduces uncertainties, the magnitude of this effect highlights the potential importance of water-stress impacts on isoprene emissions in China. This suggests a critical need for region-specific studies of emission dependence on water stress in future work.

#### 4.1.2 Evaluation against CARE-China observations

We first evaluate REAM's performance in simulating key atmospheric chemistry processes using the observations from the China National Environmental Monitoring Centre (CNEMC) Network (https://www.cnemc.cn/en/), which provides ozone (O<sub>3</sub>) and nitrogen dioxide (NO<sub>2</sub>) measurements across China. This validation step is essential before using REAM to evaluate isoprene emission inventories, as CARE-China provides only isoprene concentration measurements without broader atmospheric chemistry data. The REAM model demonstrates good agreement with observed ozone concentrations, as shown in Figs. 3 and \$10. The spatial patterns of MDA8 ozone biases are illustrated in Figs. \$11 and \$12, which show more uniform bias distributions for SieMAC models compared to MEGAN versions, particularly during high ozone episodes. The improved bias characteristics of SieMAC models during high pollution events are particularly important for air quality applications, as these episodes represent the conditions of greatest concern for human health and regulatory compliance. While modelled NO<sub>2</sub> exhibits a linear relationship with observations (Figs. 3 and \$13), there is a noted low bias that corresponds to known instrument bias issues in surface-level NO<sub>2</sub> measurements (Zhang et al., 2018; Li et al., 2019a). This general agreement in simulated ozone and NO<sub>2</sub> provides confidence in REAM's capability for simulating isoprene concentrations.

Comparing the different emission models, SieMAC shows superior performance to MEGAN simulations across most regions, as illustrated in Fig. 4. Following previous findings of distinct MEGAN performance patterns between northern and southern China by Zhang et al. (2020), we analyse model performance separately for these regions.

In northern regions including NCP, NEC, and NWC, SieMAC MR demonstrates strong consistency with observations, with 82 % and 87 % of data points falling within the evaluation criteria established by Zhang et al. (2020) (1:10 to 10:1 range), in NCP and NEC, respectively. SieMAC HR shows slightly lower but still robust performance with 78% and 82 % within range in these two regions. The inclusion of VPD effects shows great impacts on isoprene emissions in the NCP and NECP, with emissions nearly doubling when vegetation is water-stressed. However, the response of isoprene emissions to water stress exhibits nonlinear behaviour that varies with drought severity. Some studies find enhanced isoprene emissions during early or mild stages of drought due to elevated leaf temperatures from reduced stomatal conductance (Otu-Larbi et al., 2020; Kaser et al., 2022), while other studies indicate that emissions are significantly reduced under severe drought conditions (Potosnak et al., 2014; Klovenski et al., 2022). Recent modeling developments attempt to capture this complex nonlinear response, distinguishing between mild drought-induced increases and severe drought-induced decreases (Wang et al., 2022). The current VPD algorithm in SieMAC, derived from U.S. measurements, may not fully capture these nuanced responses in Chinese ecosystems. Despite these complexities, the VPD inclusion substantially improves model-measurement agreement

Formatted: Subscript

Formatted: Subscript

**Deleted:** from the China National Environmental Monitoring Centre (CNEMC) Network (https://www.cnemc.cn/en/),

Deleted: S3

Deleted: S4

Deleted: the 1:10 to 10:1 range

**Deleted:** This substantial increase not only improves modelmeasurement agreement but also highlights the critical need to investigate isoprene emissions when vegetation is water-stressed in these regions, given their potential significant impact on regional ozone chemistry. 335 and highlights the critical need to investigate isoprene emissions when vegetation is water-stresses in these regions. For SieMAC MR, the difference between measured and simulated regional averages decreases from 30 % to 3 % in NCP and from 40% to 7% in NEC when the VPD effect is included. Similarly, for SieMAC HR, the difference decreases from 43 % to 17 % in NCP and from 36 % to 5 % in NEC. This improvement reflects the model's ability to capture enhanced emissions during the dry, hot conditions typical of northern regions when the VPD effect is included. In contrast, both MEGAN models show 340 substantial underestimations, particularly MEGAN v3.1, where 58 % of predicted concentrations fall below one-tenth of observed values in NCP. The NWC region presents unique challenges, showing consistent underestimations across all models. This is primarily attributed to MODIS limitations in detecting sparse tree cover and potential bias in emission activity algorithms that were primarily developed for temperate and tropical vegetation. Recent research indicates that boreal ecosystems respond differently to environmental changes compared to plants in other climate zones (Wang et al., 2024a; Wang et al., 2024b). Despite these challenges, SieMAC still outperforms MEGAN models in this region.

In southern regions including SC, SEC, and SWC, SieMAC MR and HR achieve even stronger performance, with up to 100 % of predictions falling within the 1:10 to 10;1 range. The impact of VPD effects varies regionally, improving predictions in SWC but slightly degrading performance in SEC and SC. This variation likely stems from the relationship between VPD, temperature, and humidity. High summer temperatures in SEC and SC can elevate VPD levels and trigger emission enhancements even in humid conditions. This calls for further experiments to characterize regional plant responses to high VPD in China. MEGAN v2.1 performs reasonably well in SC and SWC but underestimates in SEC, while MEGAN v3.1 shows broader underestimation, capturing only 42 % of observations in SEC.

The distribution of model/measurement ratios in Fig. 5 further demonstrates SieMAC's advantages, particularly for SieMAC MRVPD, which exhibits the narrowest distribution, with a central tendency of unity. While MEGAN v2.1 shows reasonable performance in southern regions, its heavy left-tailed distribution for northern sites indicates systematic underestimations. MEGAN v3.1 consistently shows the broadest, left-skewed distributions, reflecting widespread underestimations across all regions. We acknowledge that the CARE-China isoprene measurements, obtained using canister sampling techniques, may have inherent uncertainties related to sampling and storage procedures (Plass-Dülmer et al., 2006), representing an additional source of uncertainty in model-measurement comparisons.

#### 360 4.1.3 Evaluation with OMI HCHO

Satellite observations of HCHO vertical columns provide an independent method for evaluating simulated isoprene emissions for regions dominated by biogenic emissions. During summer, isoprene oxidation represents the main source of HCHO over most of China's land area (Zhang et al., 2021; Fan et al., 2021; Cao et al., 2018; Wang et al., 2017a). While anthropogenic VOCs can dominate HCHO production in metropolitan regions, these urban areas comprise only a small fraction of the total land area. The spatial distribution of HCHO columns thus generally reflects isoprene emission distribution patterns across much of China (Wang et al., 2021a).

Deleted: acceptable

Deleted: range

The agreement between simulated and observed HCHO columns is examined after interpolating the OMI HCHO observations to match the REAM 36 km grid resolution. Using Pearson correlation coefficients (*r*) shown in Fig. 6, SieMAC MR and MR<sub>VPD</sub> show correlation values of 0.64 and 0.65 with OMI HCHO observations, notably higher than both MEGAN v2.1 and v3.1. Given that summer HCHO distributions are strongly influenced by isoprene oxidation, these higher correlations suggest that SieMAC MR better represents the spatial distribution of isoprene emissions. The relatively poor performance of SieMAC HR compared to MR could be attributed to the bias in its tree distributions, which strictly relies on the Vegetation Atlas.

The improvement in SieMAC is further supported by direct comparisons between modelled and observed HCHO vertical columns shown in Fig. 7. Since all model simulations use identical meteorological fields, chemical mechanisms, and emissions except for isoprene, the differences in model performance can be attributed directly to the isoprene emission estimates. The spatial correlation coefficients among all simulations are high (0.87-0.9), reflecting the contribution of isoprene emissions to high HCHO columns. MEGAN v3.1 shows a substantial underestimation of HCHO magnitudes with a large negative mean bias (MB =  $-2.2 \times 10^{15}$  molec cm<sup>2</sup>) and a high root mean square error (RMSE =  $4.35 \times 10^{15}$  molec cm<sup>2</sup>). In contrast, SieMAC configurations demonstrate much better overall performance with smaller biases and RMSE values, indicating that SieMAC not only captures spatial patterns but also reproduces the observed HCHO magnitudes more accurately. This suggests that while MEGAN v3.1 may preserve some spatial relationships, it systematically underestimates actual HCHO concentrations, whereas SieMAC provides more realistic emission estimates.

## 4.2 Spatiotemporal distributions of SieMAC Estimates

We now examine the spatiotemporal patterns of SieMAC isoprene emissions. Understanding these patterns is crucial due to isoprene's critical role in ozone formation and China's severe summer ozone pollution. We analyse the spatial distribution of isoprene emissions, present estimates for three highly urbanized and densely populated metropolitan areas with significant ozone pollution, explore the contributions of various PFTs and their temporal variations during the summer of 2013. These insights could provide valuable context for analysing isoprene's impacts on regional ozone levels.

## 4.2.1 Spatial distribution

The spatial distributions of summer isoprene emissions over China exhibit distinct patterns in different models (Fig. 8). While all models capture the general contrast between high emissions in eastern China and minimal emissions in the sparsely vegetated western regions, there are substantial differences in both magnitude and spatial patterns that call for detailed examination.

A key distinction between SieMAC HR and MR lies in their spatial continuity patterns. HR exhibits more heterogeneous distributions with abrupt transitions and isolated emission hotspots, reflecting its preservation of detailed local variations in tree species composition from the Vegetation Atlas. In contrast, MR shows more gradual spatial transitions due to its application of provincial-level tree compositions to MODIS vegetation cover data. Despite these methodological

Formatted: Superscript

Formatted: Superscript

**Deleted:** the reduced root mean square errors (RMSE) and mean biases (MB) in REAM simulations using SieMAC emissions can be attributed directly to improved isoprene emission estimates

differences, both approaches identify the Qinling mountains, encompassing southern Shaanxi, northern Hubei, and western

Henan provinces, as a major emission source, with emissions exceeding 12 nmol m<sup>-2</sup> s<sup>-1</sup> due to extensive Quercus forests.

MEGAN models identify the central southern China region, including Hunan, Jiangxi, Fujian, and southern Anhui, as the primary emission source, with higher emissions compared to the Qinling region. While MEGAN also shows elevated emissions in northern Hubei province, the high emissions do not extend to the Qinling mountains as SieMAC. MEGAN v2.1 and v3.1 share similar spatial patterns, though v3.1 consistently simulates lower emissions with peak values of around 8 nmol m<sup>-2</sup> s<sup>-1</sup> compared to approximately 12 nmol m<sup>-2</sup> s<sup>-1</sup> in v2.1, reflecting updates to emission factors and algorithms (Guenther et al., 2012; Jiang et al., 2018b; Chen et al., 2022). While SieMAC estimates align more closely with MEGAN v2.1 in magnitude, they show distinct spatial patterns and notably higher emissions across broad areas of eastern China, particularly in the Qinling region where SieMAC exceeds MEGAN v2.1 by up to 9 nmol m<sup>-2</sup> s<sup>-1</sup>. These differences primarily stem from MEGAN's use of ecoregion-based distributions versus SieMAC's incorporation of detailed Chinese vegetation data. Additionally, SieMAC HR and MR predict much higher isoprene emissions at 5.5-5.8 nmol m<sup>-2</sup> s<sup>-1</sup> over the polluted NCP region compared to MEGAN v2.1 and v3.1 at 1.4-2.8 nmol m<sup>-2</sup> s<sup>-1</sup>, indicating a larger biogenic contribution to the observed severe ozone episodes in the region than previously thought (e.g., Ma et al., 2019).

The inclusion of VPD effects in SieMAC significantly enhances emissions in regions experiencing hot and dry summers (Fig. 9). The most pronounced increases of up to 10 nmol m<sup>-2</sup> s<sup>-1</sup> occur in Shandong province and the Qinling region. These enhancements highlight the potential importance of water stress effects on regional emissions and atmospheric chemistry. However, as shown in the model evaluation against CARE-China observations and OMI HCHO data, further field measurements would help better constrain these effects across Chinese ecosystems.

#### 4.2.2 Emission Estimates for Major Megacity-cluster Areas

Table 4 lists emission estimates for the three most developed and polluted megacity-cluster regions in China: NCP, the Yangtze River Delta (YRD), and the Pearl River Delta (PRD). The spatial distribution of these study regions is shown in Fig. §14. While SieMAC predictions are generally closer to MEGAN v2.1 than MEGAN v3.1, significant differences still exist between SieMAC and MEGAN v2.1. The substantial differences between MEGAN v2.1 and v3.1 estimates also highlight the uncertainties in current isoprene emission estimates for these important megacity-cluster regions.

Previous studies have documented that isoprene emissions enhance regional ozone production in NCP and YRD.

However, studies constraining isoprene emissions in these areas are currently limited and primarily focus on subregions rather than the entire regions. SieMAC without VPD algorithm predicts isoprene fluxes of 1.16-1.26 kg C km<sup>-2</sup> h<sup>-1</sup> in NCP and 1.33-1.45 kg C km<sup>-2</sup> h<sup>-1</sup> in YRD, approximately 2.1- and 1.7-times MEGAN v2.1 estimates of 0.59 kg C km<sup>-2</sup> h<sup>-1</sup> and 0.84 kg C km<sup>-2</sup> h<sup>-1</sup>, respectively. These results suggest that isoprene's impact on regional air quality may be more significant than previously recognized.

Including VPD effects further increases emission estimates to 1.77-1.92 kg C  $km^{-2}$   $h^{-1}$  in NCP and 1.91-2.08 kg C  $km^{-2}$   $h^{-1}$  in YRD, representing increases of ~52 % and ~44 %, respectively. Given that vegetation has frequently been water-

Deleted: areas

Deleted: M

Deleted: S5

0 stressed over NCP and YRD in recent years (Xu et al., 2015), these substantial increases highlight the need for targeted research to better characterize emission dependence on water stress in Chinese ecosystems.

In contrast to NCP and YRD, PRD has more comprehensive regional emission studies available (Table 4). PRD shows the highest isoprene fluxes among the three regions, primarily due to its meteorological conditions that favour isoprene emissions. The region's dense vegetation cover, consistently high temperatures, and strong solar radiation create optimal conditions for isoprene production. While the relative difference between SieMAC and MEGAN v2.1 is smaller in PRD compared to the other two regions, SieMAC estimates are still noticeably higher than MEGAN v2.1. The PRD's humid monsoon climate presents unique challenges for modeling VPD effects. The high summer temperatures can elevate VPD levels even in humid conditions, leading to enhanced regional emissions in SieMAC, which appears to overpredict emissions compared to observations (Fig. 4). This suggests that the current VPD algorithm, developed for different climate conditions, may require regional calibration for humid environments like PRD.

#### 4.2.3 PFT Contributions and Monthly Variations

Understanding the contributions of different PFTs to isoprene emissions and their temporal variations is crucial for accurate emission modelling in China and for informing vegetation management policies. We focus on the MR and HR simulations here due to the introduced additional uncertainties by the VPD algorithm in the southern China.

Both SieMAC MR and HR simulations show that broadleaf deciduous trees are the largest contributors at 39 % and 43 %, followed by broadleaf evergreen trees at 38 % and 33 % (Fig. 10). When combined, broadleaf trees account for 76-77 % of total isoprene emissions, consistent with their known role as dominant isoprene emitters. Grasses contribute 11 % in MR and 14 % in HR simulations, while shrubs account for 7.31 % and 8.38 %. Other vegetation types make up the remainder.

These relative contributions differ somewhat from previous studies. While Li et al. (2013, 2020) similarly identified broadleaf trees as the primary contributors at 72.9 5% and 80.2 %, they found a higher relative contribution from shrubs compared to grasses. Our higher grass contributions likely reflect the updated temperature response algorithms and emission factors assigned to grasses, particularly in northwest China, where grass emissions are pronounced.

The monthly distribution of isoprene emissions shows a clear peak in July across all model simulations (Fig. 11). This temporal pattern is consistent between SieMAC MR, HR, and both versions of MEGAN, reflecting their similar approaches to calculating emission activity factors. The slight variations between SieMAC and MEGAN estimates can be attributed to SieMAC's implementation of PFT-specific LAI and updated temperature response functions for boreal grass.

## 4.3 Impacts on ozone simulations

The differences in isoprene emissions between SieMAC and MEGAN significantly influence simulated ozone distributions across China, with distinct spatial patterns emerging from different emission scenarios (Fig. 12). Compared to MEGAN v2.1 simulations, both SieMAC MR and HR predict widespread ozone increases across eastern China, while minimal changes are observed in western China, consistent with the negligible isoprene emission differences in that region. In the North

NCP region, ozone concentrations increase by 5-10 ppbv, while other regions typically show increases less than 5 ppbv. This enhancement pattern aligns with the spatial distribution of elevated atmospheric oxidation capacity, as evidenced by increased HCHO concentrations across the regions (Fig. §16). The differences become even more pronounced when comparing against MEGAN v3.1, with ozone increases exceeding 15 ppbv around the NCP region, reflecting the lower isoprene emissions in MEGAN v3.1.

The incorporation of VPD effects in SieMAC further amplifies these patterns (Figs. 12 and £17). SieMAC MRvpD and HRvpD simulations show ozone increases exceeding 15 ppbv in the NCP region, while other areas rarely exceed 10 ppbv increases. The regional differences in ozone response can be attributed to variations in ozone-NO<sub>x</sub>-VOC chemical sensitivity between different areas. The heavily polluted NCP, characterized by high NO<sub>x</sub> emissions from urbanization and industrial activities, appears to operate more in a VOC-sensitive regime where ozone production has relatively high sensitivity to reactive VOCs and therefore, additional isoprene emissions directly lead to substantial ozone increase. In contrast, regions like southern Shaanxi province in the Qinling mountains do not have large NO<sub>x</sub> sources. Ozone photochemistry in these regions is NO<sub>x</sub>-sensitive where ozone formation is less responsive to VOC increases. Additionally, the distinct topographical and meteorological differences between these regions might also contribute to the varying ozone responses. The flat NCP region experiences more persistent high-pressure systems that favour ozone accumulation, while the mountainous Shaanxi terrain might promote enhanced vertical mixing that can dilute ozone responses to isoprene increases. This enhanced sensitivity to

VPD effects is particularly significant given that the NCP already experiences China's most severe ozone pollution. These findings align with previous research (Ma et al., 2019) that documented strong correlations between water-stressed conditions, enhanced isoprene emissions, and elevated ozone levels in the NCP region.

These findings demonstrate significant spatial variations in how different isoprene emission estimates affect simulated ozone distributions across China. The pronounced ozone increases in the NCP region, particularly when accounting for VPD effects, suggest that current models may underestimate the role of biogenic emissions in China's ozone pollution.

### 5 Summary and conclusions

We developed the Speciated Isoprene Emission Model with the MEGAN Algorithm for China (SieMAC), which reveals distinct spatial patterns of isoprene emissions across China compared to previous estimates. Model evaluation against both ground-based CARE-China network observations and satellite HCHO data demonstrates SieMAC's improved performance over MEGAN v2.1 and v3.1, particularly in northern China where previous versions showed systematic underestimation. When using SieMAC emissions, the chemical transport model, REAM, produces more accurate spatial distributions of both isoprene and HCHO compared to simulations using MEGAN emissions.

SieMAC estimates summer 2013 emissions between 10.22 and 14.79 Tg C, corresponding to annual emissions of approximately 15-22 Tg C. While these magnitudes align with the range of previous studies, SieMAC indicates a significantly smaller north-south emission gradient than previously recognized. This revised spatial pattern reflects in part recent land use

Deleted: S6

Deleted: Fig. S8

Deleted:

changes, especially the extensive afforestation efforts and urban greening initiatives in northern China that have introduced high-emitting tree species across the region. The resulting higher emissions in the polluted NCP and YRD regions indicate a large contribution of biogenic emissions to severe summertime ozone pollution in these regions.

The inclusion of VPD effects in SieMAC reveals potentially significant water-stress impacts on emissions, with enhancements up to 40% during summer conditions. This finding is particularly relevant for the NCP region, where water-stress induced emission increases can contribute to severe ozone pollution episodes. These results highlight the need to better understand dehydration-emission relationships in Chinese ecosystems, especially given the region's vulnerability to climate change.

Despite these improvements, several important uncertainties remain in the current model framework. The emission factors used in SieMAC, while incorporating extensive local measurements, still cannot fully capture the diversity of China's vegetation. The assigned EFs for species lacking direct measurements introduce potential biases that could be addressed through expanded measurement campaigns. Additionally, while SieMAC uses real-time vegetation datasets to adjust distributions from the Vegetation Atlas of China, the resolution limitations of current remote sensing and vegetation statistic datasets and the dated Vegetation Atlas limit the model's ability to capture recent land use changes. The anticipated release of an updated Vegetation Atlas should help address some of these limitations.

Another significant source of uncertainty lies in the activity response algorithms. SieMAC primarily adopts MEGAN's algorithms and inherits their uncertainties. The optional VPD response factor of SieMAC, derived from U.S. measurements, may not fully represent the behaviour of Chinese vegetation. Regional differences in how plants respond to environmental changes have been documented in other studies, suggesting the need for China-specific field studies and laboratory experiments to better characterize these relationships.

Our results highlight several critical implications for environmental policy and air quality management. The significant role of biogenic isoprene in modulating regional ozone pollution is particularly evident in eastern China and the NCP region, where accurate emission representation is crucial for surface ozone control. Additionally, these findings emphasize the importance of considering air quality impacts in future afforestation projects, especially in dehydration-prone regions where VPD effects can substantially enhance isoprene emissions and subsequent ozone formation. Furthermore, our results suggest that current models may underestimate the contribution of isoprene to China's ozone pollution, particularly in polluted eastern China. Looking forward, the flexible open-source framework of SieMAC, which allows for continuous improvement and regional customization as new datasets and measurements become available, make it a valuable tool for both research and policy applications.

Figure 1. Schematic workflow for generating SieMAC land cover input data. Pink boxes denote input datasets: Vegetation Atlas, Forest Resource Statistics, tree and non-tree coverage from MODIS MOD44B, EF measurements, MODIS LAI and PFT products. Grey panels represent two processing approaches: high-resolution (HR, left) and medium-resolution (MR, right). Within each panel, orange rectangles indicate area-aggregation or scaling operations, blue rectangles indicate derivation of provincial or gridded composition/PFT fractions, and orange diamonds mark decision branches. Green boxes show the final products used as SieMAC inputs: gridded PFT fractions, EFs, and LAI.

Figure 2. Geographical distribution of the 19 CARE-China monitoring sites analysed in this study. Sites are colour-coded by region following Zhang et al. (2020): North China Plain (NCP, blue squares), Northeast China Plain (NECP, light-green squares), Northwest China (NWC, orange squares), Southeast China (SEC, magenta diamonds), South China (SC, pink diamonds), and Southwest China (SWC, grey diamonds). Provincial boundaries are shown for reference.

Figure 3. Evaluation of REAM-SieMAC MR with surface observations from the China National Environmental Monitoring Centre (CNEMC). Hourly model outputs and measurements were processed to seasonal means at each CNEMC site before comparison. (a)

Maximum daily 8-hour average ozone (MDA8); (b) daily mean NO<sub>2</sub>. Each dot represents a single site—season mean. The dashed line indicates the 1:1 relation; statistics in the lower-right corner give the root-mean-square error (RMSE), mean bias (MB), and Pearson correlation coefficient (r).

Figure 4. Comparison between simulated and measured isoprene concentrations at CARE-China sites. Northern and southern regions are evaluated separately because MEGAN exhibits region-dependent performance (Zhang et al., 2020). Panels (a-f) show results for northern sites—North China Plain (NCP, blue), Northeast China Plain (NECP, light green), and Northwest China (NWC, orange)—using six emission schemes: SieMAC MR, SieMAC HR, MEGAN v2.1, SieMAC MR<sub>VPD</sub>, SieMAC HR<sub>VPD</sub>, and MEGAN v3.1. Panels (g-l) present the same schemes for southern sites—Southeast China (SEC, magenta), South China (SC, pink), and Southwest China (SWC, dark green). Each symbol represents an observation point, while stars denote regional geometric means. The geometric mean is used here because the concentration values span multiple orders of magnitude. Both axes are logarithmic; the solid black line shows the 1:1 reference, and red dashed lines mark factors of ten (10:1 and 1:10). Percentages in the upper-left corners indicate the proportion of data points falling within the factor-of-ten lines.

Deleted:

Figure 5. Distributions of the model-to-measurement ratio for surface isoprene at CARE-China sites. Histograms are shown separately for northern (N, panels a–h) and southern (S, panels i–p) regions. Within each region, the four columns, from left to right, correspond to the SieMAC configurations: MR, MR<sub>VPD</sub>, HR, and HR<sub>VPD</sub>. Red bins represent SieMAC results, blue bins represent MEGAN v2.1, and vellow bins represent MEGAN v3.1. The x-axis is logarithmic with a bin width of 0.5. The vertical dashed line marks the 1:1 ratio.

Deleted: green

Figure 6. Spatial correspondence between modelled isoprene emissions and satellite formaldehyde (HCHO). Bars give the Pearson correlation coefficient (r) between grid-cell seasonal mean isoprene emission from each inventory—SieMAC MR, SieMAC MR<sub>VPD</sub>, SieMAC HR, SieMAC HR<sub>VPD</sub>, MEGAN v2.1, and MEGAN v3.1—and Ozone Monitoring Instrument (OMI) HCHO vertical column over mainland China for summer 2013. Higher values of r indicate a closer match in the spatial patterns of isoprene emissions and observed formaldehyde.

Figure 7. Seasonal mean of formaldehyde (HCHO) vertical columns (VC) for summer 2013 simulated with six emission inventories:

(a) SieMAC MR, (b) SieMAC HR, (c) MEGAN v2.1, (d) SieMAC MR<sub>VPD</sub>, (e) SieMAC HR<sub>VPD</sub>, and (f) MEGAN v3.1. Values are expressed in 10<sup>15</sup> molec cm<sup>2</sup>. Statistics in the lower-left corner of each panel give the mean bias (MB), the root-mean-square error (RMSE), and Pearson spatial correlation coefficient (r) between model and OMI HCHO VCs across all grid cells, quantifying the overall amplitude and spatial agreement with observations.

Deleted: <sup>3</sup>
Deleted: relative to the

Figure 8. Summertime (2013) isoprene emissions over mainland China derived from six emission schemes: (a) SieMAC MR, (b) SieMAC HR, (c) MEGAN v2.1, (d) SieMAC MR<sub>VPD</sub>, (e) SieMAC HR<sub>VPD</sub>, and (f) MEGAN v3.1. Shade shows emission rate in nmol m<sup>-2</sup> s<sup>-1</sup> (colour scale at right).

Figure 9. Spatial differences in isoprene emissions (summer 2013) among different schemes. (a) and (b) map the difference between SieMAC and MEGAN v2.1 for MR and HR, respectively (SieMAC – MEGAN v2.1). (c) and (d) map the difference between SieMAC and MEGAN v3.1 for MR and HR, respectively (SieMAC – MEGAN v3.1). (c) and (f) quantify the impact of vapour pressure deficit stress by subtracting the unstressed SieMAC fields from their VPD-enabled counterparts (SieMAC MR<sub>VPD</sub> – SieMAC MR and SieMAC HR<sub>VPD</sub> – SieMAC HR). Colours denote the magnitude of the difference in nmol m<sup>-2</sup> s<sup>-1</sup> (scale at right); red shades indicate higher emissions in the first-listed inventory, while blue shades indicate lower emissions.

Deleted: c

Deleted: d

Figure 10. Relative contribution of each PFT to the total isoprene emissions during summer 2013, shown separately for MR (a) and HR (b). BrDe Tree and BrEv Tree refer to broadleaf deciduous and broadleaf evergreen trees, respectively; "Others" comprises needleleaf trees and crops. Values next to each sector give the percentage contribution to the total national emissions attributable to that PFT.

Formatted: English (UK)

Deleted: green
Deleted: red
Formatted: Font: 9 pt
Formatted: Font: 9 pt

**Deleted:** For each inventory, the mean emission in a given month is expressed as a percentage of that inventory's total June-August emission, permitting direct comparison of seasonal progression across inventories. ¶

Formatted: Font: 9 pt

Formatted: Font: 9 pt

Figure 12. Differences in seasonal mean of daily maximum 8-hour average ozone (MDA8) between SieMAC and MEGAN simulations (summer 2013). Panels show the differences between various SieMAC configurations and MEGAN inventories: (a-d) SieMAC versus MEGAN v2.1, and (e-h) SieMAC versus MEGAN v3.1. From left to right, columns represent SieMAC MR, SieMAC HR, SieMAC MR<sub>VPD</sub>, and SieMAC HR<sub>VPD</sub> configurations. Colours indicate concentration differences in ppbv (scale at right); red shades show higher ozone concentrations in SieMAC simulations, while blue shades indicate lower concentrations.

Formatted: Font: Bold, English (UK)

Formatted: Normal, Line spacing: single

Table 1. PFT scheme used in this study

| PFT Number | Description               |  |  |
|------------|---------------------------|--|--|
| 1          | Broadleaf Evergreen Tree  |  |  |
| 2          | Broadleaf Deciduous Tree  |  |  |
| 3          | Needleleaf Evergreen Tree |  |  |
| 4          | Needleleaf Deciduous Tree |  |  |
| 5          | Shrub                     |  |  |
| 6          | Crop                      |  |  |
| 7          | Grass                     |  |  |

## 690 Table 2. REAM Simulations

| Case                     | Isoprene Emission Model                                |  |  |
|--------------------------|--------------------------------------------------------|--|--|
| SieMAC MR                | SieMAC with MR land cover, omit $\gamma_{VPD}$         |  |  |
| SieMAC HR                | SieMAC with HR land cover inputs, omit $\gamma_{VPD}$  |  |  |
| SieMAC MR <sub>VPD</sub> | Same a SieMAC MR, but includes $\gamma_{VPD}$          |  |  |
| SieMAC HR <sub>VPD</sub> | SieMAC with HR land cover, but includes $\gamma_{VPD}$ |  |  |
| MEGAN v2.1               | offline MEGAN v2.1                                     |  |  |
| MEGAN v3.1               | offline MEGAN v3.1                                     |  |  |

Table 3. Isoprene Emission Estimates for China (Unit: TgC)

| Data Source Model    |  | Study Period | Emissions (Tg C)           |
|----------------------|--|--------------|----------------------------|
| This Study SieMAC MR |  | 2013 summer  | 10.92 season <sup>-1</sup> |

| This Study              | SieMAC HR                          | 2013 summer      | 11.37 season <sup>-1</sup>   |
|-------------------------|------------------------------------|------------------|------------------------------|
| This Study              | SieMAC MR <sub>VPD</sub>           | 2013 summer      | 15.27 season <sup>-1</sup>   |
| This Study              | SieMAC HR <sub>VPD</sub>           | 2013 summer      | 15.83 season <sup>-1</sup>   |
| This Study              | MEGAN v2.1                         | 2013 summer      | 8.05 season <sup>-1</sup>    |
| This Study              | MEGAN v3.1                         | 2013 summer      | 4.47 season <sup>-1</sup>    |
| Wang et al. (2021b)     | MEGAN v2.1                         | 2011 summer      | 5.8 season <sup>-1</sup>     |
| Chen et al. (2022)      | MEGAN v2.1                         | 2014 July        | 2.00 mon <sup>-1</sup>       |
| Wang et al. (2007)      | Guenther et al. 1995               | 1999 July        | 0.95 mon <sup>-1</sup>       |
| Li et al. (2023)        | MEGAN v3.1                         | 2020             | 7.23 yr <sup>-1</sup>        |
| Ma et al. (2022)        | MEGAN v2.1                         | 2015-2019        | 13.88-14.29 yr <sup>-1</sup> |
| Wang et al. (2021a)     | MEGAN v2.1                         | 2001-2016        | 14.06 yr <sup>-1</sup>       |
| Li et al. (2020c)       | MEGAN v2.1                         | 2008, 2013, 2018 | 28.23-37.45 yr <sup>-1</sup> |
| Wang et al. (2020)      | MEGAN v2.1                         | 2001-2016        | 7.56 yr <sup>-1</sup>        |
| Wu et al. (2020)        | MEGAN v2.1                         | 2017             | 13.3 yr <sup>-1</sup>        |
| Gao et al. (2019)       | MEGAN v3.0                         | 2005-2016        | 6.13 yr <sup>-1</sup>        |
| Li and Xie (2014)       | MEGAN v2.1                         | 1999-2003        | 27.09 yr <sup>-1</sup>       |
| Stavrakou et al. (2014) | MEGAN v2.04                        | 2005             | 9.30 yr <sup>-1</sup>        |
| Li et al. (2013)        | MEGAN v2.1                         | 2003             | 23.42 yr <sup>-1</sup>       |
| Chi and Xie (2012)      | Guenther et al. 1995               | 2003             | 7.45 yr <sup>-1</sup>        |
| Fu and Liao (2012)      | MEGAN module embedded in GEOS-CHEM | 2001-2006        | 9.59 yr <sup>-1</sup>        |

| Li et al. (2012)       | PCEEA in Guenther et al.2006 | 2006 | 9.36 yr <sup>-1</sup>  |
|------------------------|------------------------------|------|------------------------|
| Tie et al. (2006)      | Guenther et al. 1993         | 2004 | 7.70 yr <sup>-1</sup>  |
| Klinger et al. (2002)  | Guenther et al. 1995         | 2000 | 4.10 yr <sup>-1</sup>  |
| Guenther et al. (1995) | Guenther et al. 1995         | 1990 | 15.00 yr <sup>-1</sup> |

Table 4. Regional Isoprene Emission Estimates (Unit: kg C km<sup>-2</sup> h<sup>-1</sup>)

| Region | Data Source         | Model                    | Study Period | Emissions<br>(kg C km <sup>-2</sup> h <sup>-1</sup> ) | Note if for subregions |
|--------|---------------------|--------------------------|--------------|-------------------------------------------------------|------------------------|
| NCP    | This Study          | SieMAC MR                | 2013 summer  | 1.26                                                  |                        |
|        |                     | SieMAC HR                | 2013 summer  | 1.16                                                  |                        |
|        |                     | SieMAC MR <sub>VPD</sub> | 2013 summer  | 1.92                                                  |                        |
|        |                     | SieMAC HR <sub>VPD</sub> | 2013 summer  | 1.77                                                  |                        |
|        |                     | MEGAN v2.1               | 2013 summer  | 0.59                                                  |                        |
|        |                     | MEGAN v3.1               | 2013 summer  | 0.29                                                  |                        |
|        | Wang et al. (2018)  | MEGAN v2.1               | 2018         | 0.18                                                  | Beijing                |
|        | Wang et al. (2003a) | GloBEIS                  | 1998 summer  | 0.17                                                  | Beijing                |
| YRD    | This Study          | SieMAC MR                | 2013 summer  | 1.33                                                  |                        |
|        |                     | SieMAC HR                | 2013 summer  | 1.45                                                  |                        |
|        |                     | SieMAC MR <sub>VPD</sub> | 2013 summer  | 1.91                                                  |                        |
|        |                     | SieMAC HR <sub>VPD</sub> | 2013 summer  | 2.08                                                  |                        |
|        |                     | MEGAN v2.1               | 2013 summer  | 0.84                                                  |                        |
|        |                     | MEGAN v3.1               | 2013 summer  | 0.48                                                  |                        |

Formatted: Indent: Left: -0.18", First line: 0.18"

|     | Lou et al. (2023)   | MEGAN v2.1               | 2020 Aug    | 1.6  | Zhejiang province |
|-----|---------------------|--------------------------|-------------|------|-------------------|
| PRD | This Study          | SieMAC MR                | 2013 summer | 1.46 |                   |
|     |                     | SieMAC HR                | 2013 summer | 1.75 |                   |
|     |                     | SieMAC MR <sub>VPD</sub> | 2013 summer | 1.70 |                   |
|     |                     | SieMAC HR <sub>VPD</sub> | 2013 summer | 2.04 |                   |
|     |                     | MEGAN v2.1               | 2013 summer | 0.78 |                   |
|     |                     | MEGAN v3.1               | 2013 summer | 0.53 |                   |
|     | Situ et al. (2014)  | MEGAN v2.1               | 2008 Fall   | 2.4  | Dinghu Mountain   |
|     | Wang et al. (2011b) | MEGAN v2.0               | 2003 summer | 0.42 |                   |
|     | Zheng et al. (2010) | GloBEIS                  | 2006        | 0.16 |                   |

## 695 Data and Code Availability

The current version of SieMAC standalone code is available on the following GitHub link: https://github.com/Cathiiie/SieMAC\_Gamma under the MIT licence. The exact version of the model described in this paper is archived on Zenodo under DOI: 10.5281/zenodo.15740701 (Xi, 2025). Setup instructions and execution steps are outlined in the README file. The sample input files, including PFT-specific emission factors, LAI datasets, sample meteorological inputs, and benchmark outputs, are available at http://apollo.eas.gatech.edu/data/. Complete emission inventory outputs, additional meteorological data files, and supplementary datasets can be provided upon request.

## **Author contribution**

YHW and SJX designed the study. SJX developed the model, performed analysis, and wrote the paper, with contribution from all co-authors.

## 705 Competing interests

The authors have no conflicting interests to share.

**Deleted:** Setup instructions and execution steps are outlined in the README file. Sample input files are provided at http://apollo.cas.gatech.edu, but users are encouraged to generate custom inputs representative their own study regions and period.

## Acknowledgements

This work is supported in part by the National Science Foundation Atmospheric Chemistry Program (Grant 1743401). XY is supported by the National Key R&D Program of China (No.2023YFC3706202) and the National Natural Science Foundation of China (No. 42375111). Model simulations and analysis were supported by the high-performance computing resources provided by PACE (Partnership for an Advanced Computing Environment) at the Georgia Institute of Technology, Atlanta, Georgia, USA.

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
