# Peer review of "The Speciated Isoprene Emission Model with the MEGAN Algorithm for China (SieMAC)"

_EGUsphere, 2025_

## Referee Comment (RC1)

A recommended figure to show in the paper, the mean biases in MDA8 $O_3$ with a threshold of 60 ppb.

---

## Author Comment (AC1)

[Figure]

**Figure 1. Spatial distribution of MDA8 O₃ mean bias for summer 2013 (all days).** Mean bias (model minus observation) at each CNEMC monitoring site for six emission inventories: (a) SieMAC MR, (b) SieMAC HR, (c) MEGAN v2.1, (d) SieMAC MRVPD, (e) SieMAC HRVPD, and (f) MEGAN v3.1. Colors indicate bias magnitude in ppbv (scale at right); red shades show model overestimation and blue shades show model underestimation relative to observations.

[Figure]

**Figure 2. Spatial distribution of MDA8 O$_3$ mean bias for high ozone days (MDA8 O$_3$ ≥ 60 ppbv).** Mean bias (model minus observation) at each CNEMC monitoring site during days when observed MDA8 ozone concentrations exceed 60 ppbv, shown for six emission inventories: (a) SieMAC MR, (b) SieMAC HR, (c) MEGAN v2.1, (d) SieMAC MRVPD, (e) SieMAC HRVPD, and (f) MEGAN v3.1. Colors indicate bias magnitude in ppbv (scale at right); red shades show model overestimation and blue shades show model underestimation relative to observations.

---

## Author Comment (AC2)

**Added Figures:**

[Figure]

**Figure S1.** PFT-specific leaf area index (LAI) distributions of the MR approach for July 2013, representing peak summer conditions when isoprene emissions are maximum.

[Figure]

**Figure S2.** PFT-specific leaf area index (LAI) distributions of the HR approach for July 2013, representing peak summer conditions when isoprene emissions are maximum.

[Figure]

**Figure S3.** Combined emission factor and PFT fraction products (EF × fraction) for the MR approach, representing the emission potential per unit area for each vegetation type.

[Figure]

**Figure S4.** Combined emission factor and PFT fraction products (EF × fraction) for the HR approach, representing the emission potential per unit area for each vegetation type.

[Figure]

**Figure S5.** Data integration workflow for the MR approach, showing the integration of Forest Resource Statistics, Vegetation Atlas, and MODIS products.

[Figure]

**Figure S6.** Data integration workflow for the HR approach, showing the MODIS-based methodology with simplified data requirements.

[Figure]

**Figure S7.** Schematic diagram of the REAM model setup showing the integration of SieMAC isoprene emissions and evaluation against observational datasets.

**Revised Figures:**

[Figure]

**Revised Figure 9.** Spatial differences in isoprene emissions (summer 2013) among different schemes. (a) and (b) map the difference between SieMAC and MEGAN v2.1 for MR and HR, respectively (SieMAC – MEGAN v2.1). (c) and (d) map the difference between SieMAC and MEGAN v3.1 for MR and HR, respectively (SieMAC – MEGAN v3.1). (e) and (f) quantify the impact of vapour pressure deficit stress by subtracting the unstressed SieMAC fields from their VPD-enabled counterparts (SieMAC MRVPD – SieMAC MR and SieMAC HRVPD – SieMAC HR). Colours denote the magnitude of the difference in nmol m-2 s-1 (scale at right); red shades indicate higher emissions in the first-listed inventory, while blue shades indicate lower emissions.

[Figure]

**Revised Figure 11.** Monthly variation of isoprene emissions for summer 2013 in four inventories: SieMAC MR (dark blue), SieMAC HR (light blue), MEGAN v2.1 (brown), and MEGAN v3.1 (grey). (a) shows the relative monthly emissions expressed as a percentage of each inventory's total summer emission, permitting direct comparison of seasonal progression across inventories. (b) presents absolute monthly emissions in Tg C mon[-1], showing the magnitude differences between emission inventories.

[Figure]

**Revised Figure 5.** Distributions of the model-to-measurement ratio for surface isoprene at CARE-China sites. Histograms are shown separately for northern (N, panels a–h) and southern (S, panels i–p) regions. Within each region, the four columns, from left to right, correspond to the SieMAC configurations: MR, MR$_{VPD}$, HR, and HR$_{VPD}$. Red bins represent SieMAC results, blue bins represent MEGAN v2.1, and yellow bins represent MEGAN v3.1. The x-axis is logarithmic with a bin width of 0.5. The vertical dashed line marks the 1:1 ratio.

[Figure]

**Revised Figure 6.** Spatial correspondence between modelled isoprene emissions and satellite formaldehyde (HCHO). Bars give the Pearson correlation coefficient (r) between grid-cell seasonal mean isoprene emission from each inventory—SieMAC MR, SieMAC MR$_{VPD}$, SieMAC HR, SieMAC HR$_{VPD}$, MEGAN v2.1, and MEGAN v3.1—and Ozone Monitoring Instrument (OMI) HCHO vertical column over mainland China for summer 2013. Higher values of r indicate a closer match in the spatial patterns of isoprene emissions and observed formaldehyde.

[Figure]

**Revised Figure 7**. Seasonal mean of formaldehyde (HCHO) vertical columns (VC) for summer 2013 simulated with six emission inventories. (a) SieMAC MR, (b) SieMAC HR, (c) MEGAN v2.1, (d) SieMAC MR$_{VPD}$, (e) SieMAC HR$_{VPD}$, and (f) MEGAN v3.1. Values are expressed in 1015 molec cm$^{-2}$. Statistics in the lower-left corner of each panel give the mean bias (MB), the root-mean-square error (RMSE), and Pearson spatial correlation coefficient (r) between model and OMI HCHO VCs across all grid cells, quantifying the overall amplitude and spatial agreement with observations.

[Figure]

**Revised Figure 8.** Summertime (2013) isoprene emissions over mainland China derived from six emission schemes: (a) SieMAC MR, (b) SieMAC HR, (c) MEGAN v2.1, (d) SieMAC MR$_{VPD}$, (e) SieMAC HR$_{VPD}$, and (f) MEGAN v3.1. Shade shows emission rate in nmol m$^{-2}$ s$^{-1}$ (colour scale at right).

[Figure]

**Revised Figure 10.** Relative contribution of each PFT to the total isoprene emissions during summer 2013, shown separately for MR (a) and HR (b). BrDe Tree and BrEv Tree refer to broadleaf deciduous and broadleaf evergreen trees, respectively; "Others" comprises needleleaf trees and crops. Values next to each sector give the percentage contribution to the total national emissions attributable to that PFT.

---

## Author Response (AR1)

**Referee Comments #1**

Thank you for your detailed and constructive comments, which have helped improve the clarity and scientific rigor of our manuscript. We have carefully addressed each of the three suggestions as detailed below:

**1. Comment on Lines 293–295:**

The inclusion of VPD effects shows great impacts on isoprene emissions in NCP and NECP, with emissions nearly doubling when vegetation is water-stressed." While some studies have shown that isoprene emissions may increase during the early or mild stages of drought, others have reported that emissions are significantly reduced under severe drought conditions (Klovenski et al., 2022). Recent developments in isoprene emission modeling, such as Wang et al., 2022, attempt to capture this nonlinear response to water stress. The authors should discuss these contrasting findings (i.e., isoprene emissions increase or decrease during drought) more clearly and provide a more balanced explanation with additional reasoning and references supporting both perspectives.

**Response:**

Thank you for this important point about the nonlinear relationship between water stress and isoprene emissions. We agree that our original discussion focused primarily on the enhancement effects during the early and mild stages of drought captured by our VPD algorithm without adequately acknowledging the broader literature on contrasting plant response during severe drought conditions. We have revised the text to provide more balanced discussion that acknowledges both the enhancement mechanism captured by our VPD algorithm and the reduction effects documented under severe drought conditions. The revision incorporates the suggested references (Klovenski et al., 2022; Wang et al., 2022) and clarifies that our VPD algorithm specifically targets the water-stress enhancement pathway, while acknowledging the need for region-specific studies to fully capture the complex nonlinear drought-emission relationships across different stress severities.

**Revised text (Line 317-335):**

The inclusion of VPD effects shows great impacts on isoprene emissions in the NCP and NECP, with emissions nearly doubling when vegetation is water-stressed. However, the response of isoprene emissions to water stress exhibits nonlinear behavior that varies with drought severity. Some studies find enhanced isoprene emissions during early or mild stages of drought due to elevated leaf temperatures from reduced stomatal conductance (Otu-Larbi et al., 2020; Kaser et al., 2022), while other studies indicate that emissions are significantly reduced under severe drought conditions (Potosnak et al., 2014; Klovenski et al., 2022). Recent modeling developments attempt to capture this complex

nonlinear response, distinguishing between mild drought-induced increases and severe drought-induced decreases (Wang et al. 2022). The current VPD algorithm in SieMAC, derived from U.S. measurements, may not fully capture these nuanced responses in Chinese ecosystems. Despite these complexities, the VPD inclusion substantially improves model-measurement agreement and highlights the critical need to investigate isoprene emissions when vegetation is water-stresses in these regions.

**2. Comment on Figures S7 and S8:**

The authors should explain why MDA8  $O_3$  increased significantly over the NCP region, while  $O_3$  levels did not rise as much over Shaanxi province, despite higher HCHO values in the SieMAC-MRVPD scenario compared to the SieMAC-MR case in both regions. A discussion of the potential chemical or meteorological factors contributing to these regional differences would help clarify the results. Or could ozone-NOx-VOC sensitivity explain the O3 response differences (since O3 changes little in response to increasing VOC in the NOx-sensitive regime)?

**Response:**

Thank you for this insightful observation about the regional differences in ozone response. You correctly notice that despite both the NCP region and Shaanxi province showing elevated HCHO concentrations in the SieMAC-MRVPD scenario, the ozone increases differ substantially between these regions. We agree that this warrants explanation and have added discussion to clarify these regional differences.

As you suggested, the differential ozone response likely reflects differences in ozone-NOx-VOC sensitivity regimes between the two regions. This explanation aligns well with our findings. We have revised the text to explain that the heavily polluted NCP region operates primarily in a VOC-limited regime where additional isoprene directly enhances ozone formation, while Shaanxi's lower NOx environment may experience mixed sensitivity conditions. We also discuss how the distinct topographical and meteorological differences might contribute to these varying ozone responses.

**Revised Section 4.3, paragraph 2 (Added discussions, line 479 - 487):**

The incorporation of VPD effects in SieMAC further amplifies these patterns (Fig. 12 and Fig. S17). SieMAC MRVPD and HRVPD simulations show ozone increases exceeding 15 ppbv in the NCP region, while other areas rarely exceed 10 ppbv increases. The regional differences in ozone response can be attributed to variations in ozone-NOx-VOC chemical sensitivity between different areas. The heavily polluted NCP, characterized by high NOx emissions from urbanization and industrial activities, appears to operate more in a VOC-sensitive regime where ozone production has relatively high sensitivity to reactive VOCs and therefore, additional isoprene emissions directly lead to substantial ozone increase. In

contrast, regions like southern Shaanxi province in the Qinling mountains do not have large NOx sources. Ozone photochemistry in these regions is NOx-sensitive where ozone formation is less responsive to VOC increases. Additionally, the distinct topographical and meteorological differences between these regions might also contribute to the varying ozone responses. The flat NCP region experiences more persistent high-pressure systems that favour ozone accumulation, while the mountainous Shaanxi terrain might promote enhanced vertical mixing that can dilute ozone responses to isoprene increases. This enhanced sensitivity to VPD effects is particularly significant given that the NCP already experiences China's most severe ozone pollution. These findings align with previous research (Ma et al., 2019) that documented strong correlations between water-stressed conditions, enhanced isoprene emissions, and elevated ozone levels in the NCP region.

**3. Comment on Figure 12:**

It would be helpful if the authors could include additional figures showing the spatial distribution of MDA8  $O_3$  mean bias compared to observations for each model (as illustrated in the attached figure), particularly for high  $O_3$  days (MDA8  $O_3 \ge 60$  ppb). These spatial patterns of mean bias could provide valuable insights into whether  $O_3$  are being overestimated or underestimated in specific regions.

**Response:**

Thank you for this excellent suggestion. We have created the requested figures showing the spatial distribution of MDA8  $O_3$  mean bias compared to CNEMC observations for each model (Figures 1 and 2). We provide two sets of maps: one for all summer days and another specifically for high ozone days with MDA8  $O_3 \ge 60$  ppb, as you suggested. We have also added discussion of these spatial bias patterns to Section 4.1.1, which reveals that SieMAC models correct the low biases of MEGAN versions in eastern China, particularly during high ozone episodes when accurate simulation is most critical for air quality applications.

Added Figure:

Figure S11. Spatial distribution of MDA8 O3 mean bias for summer 2013 (all days). Mean bias (model minus observation) at each CNEMC monitoring site for six emission inventories: (a) SieMAC MR, (b) SieMAC HR, (c) MEGAN v2.1, (d) SieMAC MRVPD, (e) SieMAC HRVPD, and (f) MEGAN v3.1. Colors indicate bias magnitude in ppbv (scale at right); red shades show model overestimation and blue shades show model underestimation relative to observations.

Figure S12. Spatial distribution of MDA8 O3 mean bias for high ozone days (MDA8 ≥ 60 ppbv). Mean bias (model minus observation) at each CNEMC monitoring site during days when observed MDA8 ozone concentrations exceed 60 ppbv, shown for six emission inventories: (a) SieMAC MR, (b) SieMAC HR, (c) MEGAN v2.1, (d) SieMAC MRVPD, (e) SieMAC HRVPD, and (f) MEGAN v3.1. Colors indicate bias magnitude in ppbv (scale at right); red shades show model overestimation and blue shades show model underestimation relative to observations.

**Revised Section 4.1.2, paragraph 1 (Added discussions, line 303 - 306):**

The REAM model demonstrates good agreement with observed ozone concentrations from the China National Environmental Monitoring Centre (CNEMC) Network (https://www.cnemc.cn/en/), as shown in Figs. 3 and S10. The spatial patterns of MDA8 ozone biases are illustrated in Figs. S11 and S12, which show that SieMAC models correct the low biases of MEGAN versions in eastern China, particularly during high ozone episodes. The improved bias characteristics of SieMAC models during high pollution events are particularly important for air quality applications, as these episodes represent the conditions of greatest concern for human health and regulatory compliance.

**References:**

Potosnak et al. (2014). Observed and modeled ecosystem isoprene fluxes from an oak-dominated temperate forest and the influence of drought stress. Atmospheric Environment, 84, 314–322. https://doi.org/10.1016/j.atmosenv.2013.11.055.

Otu-Larbi et al. (2020). Modelling the effect of the 2018 summer heatwave and drought on isoprene emissions in a UK woodland. Global Change Biology, 26, 2320–2335. https://doi.org/10.1111/gcb.14963.

Kaser et al. (2022). Interannual variability of terpenoid emissions in an alpine city. Atmospheric Chemistry and Physics, 22(8), 5603–5618. https://doi.org/10.5194/acp-22-5603-2022.

Klovenski et al., (2022). Interactive biogenic emissions and drought stress effects on atmospheric composition in NASA GISS ModelE. Atmospheric Chemistry and Physics, 22, 13303–13323. https://doi.org/10.5194/acp-22-13303-2022.

Wang et al. (2022). Modeling isoprene emission response to drought and heatwaves within MEGAN using evapotranspiration data and by coupling with the community land model. Journal of Advances in Modeling EarthSystems, 14, e2022MS003174. https://doi.org/10.1029/2022MS003174.

Ma et al. (2019). Substantial ozone enhancement over the North China Plain from increased biogenic emissions due to heat waves and land cover in summer 2017. Atmospheric Chemistry and Physics.19, 12195-12207, 10.5194/acp-19-12195-2019.

**Referee Comments #2**

We sincerely thank you for your thorough and constructive review. We appreciate the positive assessment of our work's aims and the recognition that SieMAC represents "an important step towards mapping isoprene emissions over China" with "worthwhile contribution to our understanding of BVOC emissions". The detailed feedback will significantly improve our manuscript and our response to your detailed comments are listed below:

**1. General comments**

I have one major general comment. The presented model builds on the integration of two different vegetation maps to derive PFT-specific maps of LAI and emission factors. This sounds like a very difficult undertaking, it sounds very interesting, and it appears to be the major innovating step in the presented methodology. I was therefore a little disappointed not to see any of the results from this work, i.e., maps of PFT-specific LAI, and the so-called PFT fractions and PFT emission factors. Given that this is a submission to GMD, this is a presentation of a model, and this appears to be the primary innovation carried out, I would like to see a presentation of the key results from this methodological step. A lot of emphasis has been placed on demonstrating differences between SieMAC and MEGAN at the endpoints, i.e., isoprene emissions/concentrations, and ozone. A presentation of the intermediate steps would really help to advance our understanding of what changed in the underlying PFT-specific data for fraction, EF, and LAI. The authors have made attempts to explain some of these effects in the discussion in the text, but I think this would be even clearer with the use of some maps showing key findings.

Overall, the manuscript is well written, well structured, and nearly all relevant methodological steps are explained, but there are some exceptions that need to be addressed. I have detailed these below in the specific comments section.

**Response:**

Thank you for this insightful comment highlighting the importance of presenting our methodological innovations more clearly. You are absolutely correct that the integration of vegetation maps to derive PFT-specific parameters represents the core innovation of SieMAC, and we agree that showing these intermediate results would significantly enhance understanding of our approach. We have added four new supplementary figures (Figures S1-S4) that directly address this concern. We present July LAI distributions as representative of peak summer conditions, given that July shows maximum isoprene emissions across all model configurations, and spatial LAI patterns remain consistent throughout the summer months with only magnitude variations. These intermediate results directly explain the emission differences shown in the main figures. For example, the higher emissions in southern China result from the combination of high PFT-specific LAI values for broadleaf trees and their updated emission factors, while the regional differences between SieMAC and MEGAN stem from our improved representation of mixed vegetation areas and the incorporation of recent afforestation patterns. We have added

references to these figures in the main text when discussing why SieMAC outperforms MEGAN versions. We believe these additions effectively demonstrate the "what changed" aspect, showing how our methodological innovations in vegetation characterization translate to improved emission estimates across China's diverse ecosystems.

**Figures (supplement: Figures S5-S8, main text: line 216-217):**

**Figure S5.** PFT-specific leaf area index (LAI) distributions of the MR approach for July 2013, representing peak summer conditions when isoprene emissions are maximum.

**Figure S6.** PFT-specific leaf area index (LAI) distributions of the HR approach for July 2013, representing peak summer conditions when isoprene emissions are maximum.

**Figure S7.** Combined emission factor and PFT fraction products (EF  $\times$  fraction) for the MR approach, representing the emission potential per unit area for each vegetation type.

**Figure S8.** Combined emission factor and PFT fraction products (EF  $\times$  fraction) for the HR approach, representing the emission potential per unit area for each vegetation type.

**2. Specific comments**

**#1 Comment:**

The abstract makes no mention of which time period is presented in the paper. I think this detail should be added.

**Response:**

We agree this is an important omission. We have revised the last sentence of the abstract to include the study period. The abstract now specifies that our analysis focuses on "summer 2013 (June-August)" to provide a clear temporal context for readers.

**Revised abstract:**

Isoprene is the dominant non-methane Volatile Organic Compound (VOC) emitted from terrestrial ecosystems and plays an important role in ozone chemistry. Understanding isoprene emissions is critical for controlling air pollution. The Model of Emissions of Gases and Aerosols from Nature (MEGAN) is widely used to calculate biogenic isoprene emissions worldwide. While MEGAN predictions are good for many regions, a previous analysis of isoprene observations around China showed large discrepancies between observed and simulated isoprene concentrations. The uncertainties of isoprene emissions in China are also reflected in the large differences between MEGAN version 2.1 and 3.1. In this work, bottom-up high-resolution vegetation distributions and updated emission factors are combined with satellite data in the Speciated Isoprene Emission Model with the MEGAN Algorithm for China (SieMAC) to improve isoprene emission estimates in China. The results from this new emission inventory for summer 2013 improve upon MEGAN versions 2.1 and 3.1 when compared with isoprene observations and satellite HCHO products. This improved emission inventory is applied in a regional model, and the results indicate a potentially underestimated role of biogenic isoprene in ozone formation over polluted eastern China.

**#2 Comment:**

It is not explained why the summer of 2013 was selected for study as compared to any other time period. The authors should give some explanation for this. I am not suggesting that such a justification does not exist, but it seems a little arbitrary given that 2013 is now over 10 years ago.

**Response:**

Thank you for this important clarification request. We selected summer 2013 based on observational data availability for model evaluation. The CARE-China campaign operated from March 2012 to April 2014, providing complete summer isoprene observations for both 2012 and 2013. However, the China National Environmental Monitoring Centre (CNEMC) network, which provides essential ozone and other pollutant measurements for our model validation, began operations in 2013. Therefore, 2013 represents the first year with simultaneous availability of both comprehensive isoprene measurements (CARE-China) and key air quality observations (CNEMC ozone data), making it optimal for validating both our emission estimates and their atmospheric chemistry impacts. We also add a sentence to the end of the first paragraph in Section 3 for clarification (Zhang et al. 2020, Bai et al. 2020).

**Added Sentence in line 230-232:**

We selected summer 2013 for this study as it represents the first year with concurrent availability of CARE-China isoprene measurements and the China National Environmental Monitoring Centre (CNEMC) Network air quality observations, enabling comprehensive model evaluation (Zhang et al. 2020, Bai et al. 2020).

**#3 Comment:**

Section 2.2.1 contains a lot of details buried in a very dense, and quite difficult-to-follow, text. I like the fact that the authors have used a flow diagram in Figure 1. I do, however, think

there is a need for an additional schematic to visually demonstrate the regridding and what all of the exceptions and examples mean. There are a lot of exceptions and ifs and buts, but this makes it hard to keep track of each unique case. I would recommend adding a schematic defining these different cases in some visual sense.

**Response:**

We agree that Section 2.2.1 contains complex methodological details that would benefit from clearer visual presentation. We have added two new supplementary figures (Figures S5 and S6) that illustrate the data integration workflows for both HR and MR approaches. Figure S5 shows the HR workflow with the decision tree for integrating Forest Resource Statistics and Vegetation Atlas data, while Figure S6 shows the simplified MR approach using primarily MODIS products. These schematics clearly show: (1) the input data sources and their spatial resolutions, (2) the decision criteria and processing steps, (3) the regridding workflows, and (4) the final PFT-specific outputs. We have also revised Section 2.2.1 to reference these supplementary schematics at key points, making the complex methodology easier to follow.

**Figure S2.** Data integration workflow for the MR approach, showing the integration of Forest Resource Statistics, Vegetation Atlas, and MODIS products.

**Figure S3.** Data integration workflow for the HR approach, showing the MODIS-based methodology with simplified data requirements.

**#4 Comment:**

I would recommend restructuring Section 3 to add a new 3.1. The text prior to the current Sect. 3.1 could itself be in its own section titled Model Setup, or something like it.

**Response:**

We agree this would improve the manuscript structure and readability. We have restructured Section 3 by creating a new Section 3.1 titled "Model Setup" that contains the description of the REAM model configuration, meteorological inputs, and SieMAC implementation details. The previous Section 3.1 "CARE-China Observations" has been renumbered as Section 3.2, with all subsequent sections renumbered accordingly.

**#5 Comment:**

I would recommend adding a schematic figure to help describe the model setup described in Sect. 3.

**Response:**

Thank you for pointing this out. We agree that a schematic would enhance understanding of the model setup. We have added a new supplementary figure (Figure S7) that illustrates the

REAM model configuration and SieMAC integration. This schematic shows: ((1) the input data sources including meteorological fields from WRF and land cover datasets, (2) the parallel processing through both MEGAN and SieMAC algorithms, (3) the REAM chemical transport model setup, and (4) the model outputs used for comparison with observational datasets. We have added a reference to this figure in the new Section 3.1 Model Setup.

**Figure S9.** Schematic diagram of the REAM model setup showing the integration of SieMAC isoprene emissions and evaluation against observational datasets.

**# 6 Comment:**

For the comparison between the REAM HCHO columns and the OMI HCHO column observations, it is not described whether the averaging kernel from the OMI HCHO retrieval was applied to the model HCHO columns before making the comparison. This is an important methodological step, and it should be carried out when comparing model and satellite data to avoid spurious comparisons.

**Response:**

Thank you for noticing this important methodological consideration. We acknowledge that applying the averaging kernel to model vertical profiles is the standard practice for satellite—model comparisons. In practice, air mass factor (AMF) is applied in tropospheric vertical column computation to account for the averaging kernel effect. This is particularly important for NO2 since NOx has emission sources above the boundary layer. For HCHO retrievals, on the other hand, the dependence of air mass factor (AMF) on altitude is weak and is not

considered in HCHO AMF calculations (De Smedt et al., 2018). We therefore used the standard AMF-corrected tropospheric HCHO vertical column products in our analysis.

**#7 Comment:**

There are correlation coefficients displayed in each panel of Figure 7 for each comparison between REAM and OMI HCHO. Are these correlation coefficients indicating the spatial correlation between OMI and REAM or is there a temporal component to the correlation too? This should be clarified. Furthermore, the correlation scores listed in Figure 7 are highest for MEGAN3.1 (albeit by a small margin). Can the authors discuss this given that in the discussion about the correlation between isoprene observations and REAM results (Figure 6), correlation scores were used to argue for the improvement of SieMAC compared to MEGAN.

**Response:**

Thank you for seeking clarification on these important methodological details.

We have clarified the Figure 7 caption to explicitly state that these represent spatial correlations. The correlation coefficients are calculated between seasonal mean HCHO vertical columns from model simulations and OMI observations across all grid cells over mainland China, with no temporal component involved.

We acknowledge that MEGAN v3.1 shows a slightly higher correlation coefficient in Figure 7 (r=0.90). However, we have revised our discussion to clarify that correlation alone is insufficient for comprehensive model evaluation. While MEGAN v3.1 may capture some spatial patterns, it exhibits substantial systematic underestimation of HCHO magnitudes (MB=-2.2×1015 molec cm-2, RMSE=4.35×1015 molec cm-2). In contrast, SieMAC configurations demonstrate much better overall performance with smaller biases and RMSE values, indicating that SieMAC not only captures spatial patterns but also reproduces observed HCHO magnitudes more accurately. This supports our conclusion that SieMAC provides more realistic emission estimates than MEGAN versions. We have revised Section 4.1.3 to better discuss the trade-offs between different performance metrics and clarify why bias and RMSE are crucial for evaluating emission inventory accuracy.

**Added text in line 378-385:**

The spatial correlation coefficients among all simulations are high (0.87-0.9), reflecting the contribution of isoprene emissions to high HCHO columns. MEGAN v3.1 shows a substantial underestimation of HCHO magnitudes with a large negative mean bias (MB = -2.2×1015 molec cm-2) and a high root mean square error (RMSE = 4.35×1015 molec cm-2). In contrast, SieMAC configurations demonstrate much better overall performance with smaller biases and RMSE values, indicating that SieMAC not only captures spatial patterns but also reproduces the observed HCHO magnitudes more accurately. This suggests that while

MEGAN v3.1 may preserve some spatial relationships, it systematically underestimates actual HCHO concentrations, whereas SieMAC provides more realistic emission estimates.

**Revised Figure 7 caption:**

Figure 7. Seasonal mean of formaldehyde (HCHO) vertical columns (VC) for summer 2013 simulated with six emission inventories: (a) SieMAC MR, (b) SieMAC HR, (c) MEGAN v2.1, (d) SieMAC MRVPD, (e) SieMAC HRVPD, and (f) MEGAN v3.1. Values are expressed in 1015 molec cm-2. Statistics in the lower-left corner of each panel give the mean bias (MB), the root-mean-square error (RMSE), and Pearson spatial correlation coefficient (r) between model and OMI HCHO VCs across all grid cells, quantifying the overall amplitude and spatial agreement with observations.

**#8 Comment:**

Placing Figure 3 first in the results section seems out of place. Ozone is discussed in more detail in 4.3, so it would seem more appropriate to place the results there. I realise that this ozone comparison is part of the CARE-China comparison, but I think it makes more sense to structure based on data type rather than data source.

**Response:**

Thank you for this organizational suggestion. We recognize that the rationale for Figure 3's placement may not be sufficiently clear in our current text. We have added clarifying sentences at the opening of Section 4.1.2 to better explain that Figure 3 evaluates REAM's performance using CNEMC observations of ozone (O3) and nitrogen dioxide (NO2), which serves as essential model validation before proceeding to isoprene evaluation. Since CARE-China provides only isoprene measurements without O3 or NO2 data, CNEMC represents our only source for validating REAM's atmospheric chemistry simulation performance. We have revised the text to better explain this methodological sequence: demonstrating that REAM accurately simulates key atmospheric chemical processes is necessary before using the model to evaluate isoprene emission inventories. Section 4.3 then builds upon this established model credibility to examine how different emission inventories impact ozone concentrations.

**Added sentences to the opening paragraph of Section 4.1.2 (line 298-302):**

We first evaluate REAM's performance in simulating key atmospheric chemistry processes using the observations from the China National Environmental Monitoring Centre (CNEMC) Network (https://www.cnemc.cn/en/), which provides ozone (O3) and nitrogen dioxide (NO2) measurements across China. This validation step is essential before using REAM to evaluate isoprene emission inventories, as CARE-China provides only isoprene concentration measurements without broader atmospheric chemistry data.

**#9 Comment:**

In the comparisons in Sect. 4.1.2, the authors have defined a range from 1:10 to 10:1 (ratio of model to obs and obs to model) and say that this is acceptable. I realise that in this metric that the SieMAC results are better than either of the MEGAN options, which supports their overall conclusions, but calling this range acceptable seems like a very wide (overly wide) set of goalposts. Is there a way to explain that this huge range is acceptable in terms of qualitative or quantitative outcome for modelling? Or perhaps a different word needs to be used here. I do not think the authors want to halt

**Response:**

Thank you for this important point about the terminology used for model evaluation criteria. We acknowledge that the 1:10 to 10:1 range appears quite wide. Our choice of this range was adopted from Zhang et al. (2020), who used the same evaluation criteria for isoprene model-measurement comparisons in China. This range reflects the inherent challenges in modeling isoprene concentrations, which are extremely sensitive to environmental variability, concentrations at the same location can vary by orders of magnitude between consecutive days due to changes in environmental conditions.

However, we agree that "acceptable" may not be the most appropriate term for such a wide range. We have revised the text to replace "acceptable" with "within the evaluation criteria established by Zhang et al. (2020)" to better reflect that this represents a comparison framework rather than an aspirational target. We also acknowledge that while SieMAC performs better than MEGAN versions within this framework, continued improvements in emission modeling remain an important research priority.

Revised sentences: line 314-315, line 347.

**#10 Comment:**

Can the authors please add the emissions per unit area to Table 4. They are listed in the text, but it should be in the Figure too. Without knowing the spatial extent of each of the three zones, it is difficult to understand the meaning of each emission total.

**Response:**

Thank you for this suggestion. We have added the units (kg C km-2 h-1) to the "Emissions" column header in Table 4 for improved clarity, so readers can immediately see the units without needing to refer to the caption. This makes the per-unit-area nature of the emission values more immediately apparent, allowing for direct comparison of emission intensities across the three regions independent of their spatial extents.

**#11 Comment:**

The results of a comparison between MEGAN 3.1 and SieMAC are not present in Figure 9. There is not a clear reason why this is the case and for consistency with Figure 8, I think it would be better to add it.

**Response:**

We agree adding comparison between MEGAN 3.1 and SieMAC improves consistency and completeness. We have revised Figure 9 to include spatial differences between SieMAC configurations and MEGAN v3.1 (panels c and d), parallel to the existing SieMAC vs MEGAN v2.1 comparisons (panels a and b).

**Figures:**

**Revised Figure 9.** Spatial differences in isoprene emissions (summer 2013) among different schemes. (a) and (b) map the difference between SieMAC and MEGAN v2.1 for MR and HR, respectively (SieMAC – MEGAN v2.1). (c) and (d) map the difference between SieMAC and MEGAN v3.1 for MR and HR, respectively (SieMAC – MEGAN v3.1). (e) and (f) quantify the impact of vapour pressure deficit stress by subtracting the unstressed SieMAC fields from their VPD-enabled counterparts (SieMAC MRVPD – SieMAC MR and SieMAC HRVPD – SieMAC HR). Colours denote the magnitude of the difference in nmol m-2 s-1 (scale at right); red shades indicate higher emissions in the first-listed inventory, while blue shades indicate lower emissions.

**#12 Comment:**

Can the authors please add the absolute values to Figure 11 in addition to the relative values.

**Response:**

Thank you for your suggestion. We have revised Figure 11 to include both relative and absolute values as requested. The figure now contains two panels: the top panel maintains the relative comparison (percentage of total summer emissions) which clearly shows seasonal progression patterns, while the bottom panel displays absolute monthly emissions (Tg C mon-1) that reveal the magnitude differences between inventories.

**Figures:**

**Revised Figure 11.** Monthly variation of isoprene emissions for summer 2013 in four inventories: SieMAC MR (dark blue), SieMAC HR (light blue), MEGAN v2.1 (brown), and MEGAN v3.1 (grey). (a) shows the relative monthly emissions expressed as a percentage of each inventory's total summer emission, permitting direct comparison of seasonal progression across inventories. (b) presents absolute monthly emissions in Tg C mon-1, showing the magnitude differences between emission inventories.

**#13 Comment:**

There is no discussion on measurement technique for isoprene measurements. This is a very difficult measurement to make. It sounds like samples are being taken, which I assume means these are canister-based measurements. Canister-based measurements of isoprene are subject to various known biases, and this traces back to how the canisters are handled and cleaned. Improper handling of canisters likely leads to over-estimation of isoprene due to biotic isoprene production inside the canisters in the event that the canisters are not pasteurized, for instance. We see clear differences in online versus canister-based measurement of isoprene. Issues related to data quality and how this affects the comparisons and conclusions should therefore be discussed in a meaningful way.

Plass-Dülmer, C., Schmidbauer, N., Slemr, J., Slemr, F., and D'Souza, H.: European hydrocarbon intercomparison experiment AMOHA part 4: Canister sampling of ambient air, Journal of Geophysical Research Atmospheres, 111, https://doi.org/10.1029/2005JD006351, 2006.

**Response:**

Thank you for raising this important point about measurement uncertainties. We acknowledge that isoprene measurements are technically challenging and that different measurement techniques can introduce systematic biases. The CARE-China isoprene measurements were obtained using Silonite-treated stainless steel canisters with detailed quality control procedures as described in Zhang et al. (2020). While the authors of the CARE-China dataset address measurement methodology and quality assurance protocols in their original publication, we recognize that canister-based measurements may have inherent limitations compared to online measurements.

We have added a brief acknowledgment of measurement uncertainties at the end of our result section 4.1.2 for CARE-China comparison, noting that while our model-measurement comparisons demonstrate clear improvements with SieMAC, measurement technique limitations represent an additional source of uncertainty in the evaluation. The consistent improvements of SieMAC across multiple evaluation metrics (ground-based isoprene, satellite HCHO, and atmospheric chemistry validation) provide confidence in our conclusions despite individual measurement uncertainties.

**Added text in line 357-359:**

We acknowledge that the CARE-China isoprene measurements, obtained using canister sampling techniques, may have inherent uncertainties related to sampling and storage procedures (Plass-Dülmer et al., 2006), representing an additional source of uncertainty in model-measurement comparisons.

**#14 Comment:**

There is not enough clarity provided in the data and code availability section of the manuscript. The weblink for the data takes you to a Georgia Tech page for the Wang research group. After some clicking I found my way to the data section of this webpage. But I would recommend simply listing the weblink to get directly to the data for ease of use. Furthermore, it is not clearly described if this data is for creating emissions over China to replicate the emissions presented in the manuscript, and this should be clarified. Ideally, the data provided should allow the full replication of the emission inventory. There is no mention of the REAM model code availability (and input data), but perhaps the authors consider that this is too far for readers to go. I am not sure about what the journal requires, but should it not be made clear that the REAM simulations are considered to be beyond the scope for replication and essentially is not being supported by the authors?

**Response:**

Thank you for this important feedback about data and code accessibility. We have revised the data availability section to provide more direct access and clarify the scope of available materials.

We have updated the weblink to point directly to the data repository (http://apollo.eas.gatech.edu/data/) rather than the research group homepage for easier access. The available dataset includes the PFT-specific emission factors and LAI data produced and applied in this study, along with 2-day meteorological input files and benchmark model emission outputs. This dataset enables users to replicate the SieMAC emission calculations presented in the manuscript and validate their model outputs against our benchmarks.

We acknowledge that the complete meteorological dataset is too large to host permanently online, but we are prepared to provide additional meteorological data files upon request from interested users. The full emission inventory outputs presented in this study are also available upon request. Regarding the REAM model code, we clarify that this research model's code does not currently have proper documentation for public release. However, the emission inventory methodology and input data provided are sufficient for researchers to implement SieMAC with other atmospheric chemistry models.

**Revised data availability section:**

The current version of SieMAC standalone code is available on the following GitHub link: <a href="https://github.com/Cathiiie/SieMAC\_Gamma">https://github.com/Cathiiie/SieMAC\_Gamma</a> under the MIT licence. The exact version of the model described in this paper is archived on Zenodo under DOI: 10.5281/zenodo.15740701 (Xi, 2025). Setup instructions and execution steps are outlined in the README file. The sample input files, including PFT-specific emission factors, LAI datasets, sample meteorological inputs, and benchmark outputs, are available at

http://apollo.eas.gatech.edu/data/. Complete emission inventory outputs, additional meteorological data files, and supplementary datasets can be provided upon request.

**3. Technical comments**

**#1 Comment:**

Remove plural from "compounds" and "VOCs" in the first line of the abstract.

**Response:**

Thank you for pointing this out. We have corrected this in the main text.

**#2 Comment:**

Line 36. MEGAN is the most widely used what? The sentence structure requires a definition of what MEGAN is.

**Response:**

Thank you for catching this incomplete sentence. We have revised the sentence.

Revised sentence (line 38 - 39):

The Model of Emissions of Gases and Aerosols from Nature (MEGAN) is the most widely used biogenic emission model and serves as the standard module for estimating biogenic isoprene emissions in chemistry transport models.

**#3 Comment:**

Line 37. Add "Chinese" before national. Speaking about National without a definition of which nation is imprecise.

**Response:**

Thank you for pointing this out. We revised the sentence as "However, MEGAN-based national isoprene emission estimates for China..." (*line 40*).

**#4 Comment:**

Line 76. Add the last access date to the megan weblink.

**Response:**

Thank you for pointing this out. We have added the last access date (*line 79*).

**#5 Comment:**

Line 178. It looks like there is an unaccepted document change.

**Response:**

Thank you for pointing this out. We have corrected this in the main text.

**#6 Comment:**

I think that Figures 5, 6, 7, 8, and 10 do not conform to colour-blindness friendly guidelines. Mixing Red and green in figures creates problems for most cases of dichromatic colour-blindness. Please us this indicator recommended by Copernicus to help rectify this issue:

https://www.color-blindness.com/coblis-color-blindness-simulator/

**Response:**

We appreciate this important accessibility concern. We have revised Figures 5, 6, 7, 8, and 10 to use color-blind friendly color schemes that avoid problematic red-green combinations. The new color palettes were tested using the Copernicus-recommended color-blindness simulator (https://www.color-blindness.com/coblis-color-blindness-simulator/) to ensure accessibility for readers with dichromatic color-blindness.

**Revised Figures:**

**Revised Figure 5.** Distributions of the model-to-measurement ratio for surface isoprene at CARE-China sites. Histograms are shown separately for northern (N, panels a–h) and southern (S, panels i–p) regions. Within each region, the four columns, from left to right, correspond to the SieMAC configurations: MR, MRVPD, HR, and HRVPD. Red bins represent SieMAC results, blue bins represent MEGAN v2.1, and yellow bins represent MEGAN v3.1. The x-axis is logarithmic with a bin width of 0.5. The vertical dashed line marks the 1:1 ratio.

**Revised Figure 6.** Spatial correspondence between modelled isoprene emissions and satellite formaldehyde (HCHO). Bars give the Pearson correlation coefficient (r) between grid-cell seasonal mean isoprene emission from each inventory—SieMAC MR, SieMAC MRVPD, SieMAC HR, SieMAC HRVPD, MEGAN v2.1, and MEGAN v3.1—and Ozone Monitoring Instrument (OMI) HCHO vertical column over mainland China for summer 2013. Higher values of r indicate a closer match in the spatial patterns of isoprene emissions and observed formaldehyde.

**Revised Figure 7**. Seasonal mean of formaldehyde (HCHO) vertical columns (VC) for summer 2013 simulated with six emission inventories. (a) SieMAC MR, (b) SieMAC HR, (c) MEGAN v2.1, (d) SieMAC MRVPD, (e) SieMAC HRVPD, and (f) MEGAN v3.1. Values are expressed in 1015 molec cm-2. Statistics in the lower-left corner of each panel give the mean bias (MB), the root-mean-square error (RMSE), and Pearson spatial correlation coefficient (r) between model and OMI HCHO VCs across all grid cells, quantifying the overall amplitude and spatial agreement with observations.

**Revised Figure 8.** Summertime (2013) isoprene emissions over mainland China derived from six emission schemes: (a) SieMAC MR, (b) SieMAC HR, (c) MEGAN v2.1, (d) SieMAC MRVPD, (e) SieMAC HRVPD, and (f) MEGAN v3.1. Shade shows emission rate in nmol  $m^{-2}$  s-1 (colour scale at right).

**Revised Figure 10.** Relative contribution of each PFT to the total isoprene emissions during summer 2013, shown separately for MR (a) and HR (b). BrDe Tree and BrEv Tree refer to broadleaf deciduous and broadleaf evergreen trees, respectively; "Others" comprises needleleaf trees and crops. Values next to each sector give the percentage contribution to the total national emissions attributable to that PFT.

**References:**

Zhang et al. (2020): Isoprene mixing ratios measured at twenty sites in China during 2012–2014: Comparison with model simulation. Journal of Geophysical Research: Atmospheres, 125, e2020JD033523. https://doi.org/10.1029/2020JD033523.

Bai et al. (2020): A homogenized daily in situ PM2.5 concentration dataset from the national air quality monitoring network in China, Earth Syst. Sci. Data, 12, 3067–3080, https://doi.org/10.5194/essd-12-3067-2020, 2020.

De Smedt et al. (2018): Algorithm theoretical baseline for formaldehyde retrievals from S5P TROPOMI and from the QA4ECV project, Atmos. Meas. Tech., 11, 2395–2426, https://doi.org/10.5194/amt-11-2395-2018, 2018.

Xi (2025): SieMAC Offline Code, 10.5281/zenodo.15740701, 2025.